# Conformal Language Model Reasoning with Coherent Factuality

Maxon Rubin-Toles[1*], Maya Gambhir[1*], Keshav Ramji[1,2], Aaron Roth[1], Surbhi Goel[1]

[1]University of Pennsylvania, [2]IBM Research AI
* Denotes equal contribution

## Abstract

Language models are increasingly being used in important decision pipelines, so ensuring the correctness of their outputs is crucial. Recent work has proposed evaluating the "factuality" of claims decomposed from a language model generation and applying conformal prediction techniques to filter out those claims that are not factual. This can be effective for tasks such as information retrieval, where constituent claims may be evaluated in isolation for factuality, but is not appropriate for reasoning tasks, as steps of a logical argument can be evaluated for correctness only within the context of the claims that precede them. To capture this, we define "coherent factuality" and develop a conformal-prediction-based method to guarantee coherent factuality for language model outputs. Our approach applies split conformal prediction to subgraphs within a "deducibility" graph that represents the steps of a reasoning problem. We evaluate our method on mathematical reasoning problems from the MATH and FELM datasets and find that our algorithm consistently produces correct and substantiated orderings of claims, achieving coherent factuality across target coverage levels.[1] Moreover, we achieve 90% factuality on our stricter definition while retaining 80% or more of the original claims, highlighting the utility of our deducibility-graph-guided approach.

## 1 Introduction

As foundation models become ubiquitous, it is important to verify the correctness of their generations. Consequently, ensuring the factuality and reliability of the outputs of these models is an area of active and growing research. One line of research (Guo et al., 2017; Jiang et al., 2021; Lin et al., 2022; Mielke et al., 2022; Detommaso et al., 2024; Ahdritz et al., 2024) attempts to catch errors by quantifying model uncertainty; however, these methods are often difficult to apply in practical settings where output spaces are intractably large and uncertainty signals, like logit weights, are not accessible for many proprietary models.

Recently, conformal prediction has been explored as an uncertainty quantification technique to address correctness in language model (LM) outputs. In particular, (Mohri & Hashimoto, 2024) apply split conformal prediction to filter generations by removing weak claims according to some threshold calibrated to a desired error rate $\alpha$. Subsequent work (Cherian et al., 2024) issues weaker but adaptive guarantees to ensure output completeness. However, both works implicitly assume the factuality of a claim can be independently evaluated, which limits their generalizability to reasoning domains, where claims require substantiation. For example, in solving math problems, a given step is often deduced as a result of preceding steps: generally, logical arguments require substantiation.

To tackle this challenge, we propose a new notion of factuality to account for the structure of reasoning problems, provide an algorithm which applies split conformal prediction to filter claims over a graph representation, and give correctness guarantees over the filtered output:

**A well-defined notion of coherent factuality.** We present a notion of factuality which accounts for inter-claim dependence to evaluate correctness in a more faithful manner. This definition requires

---

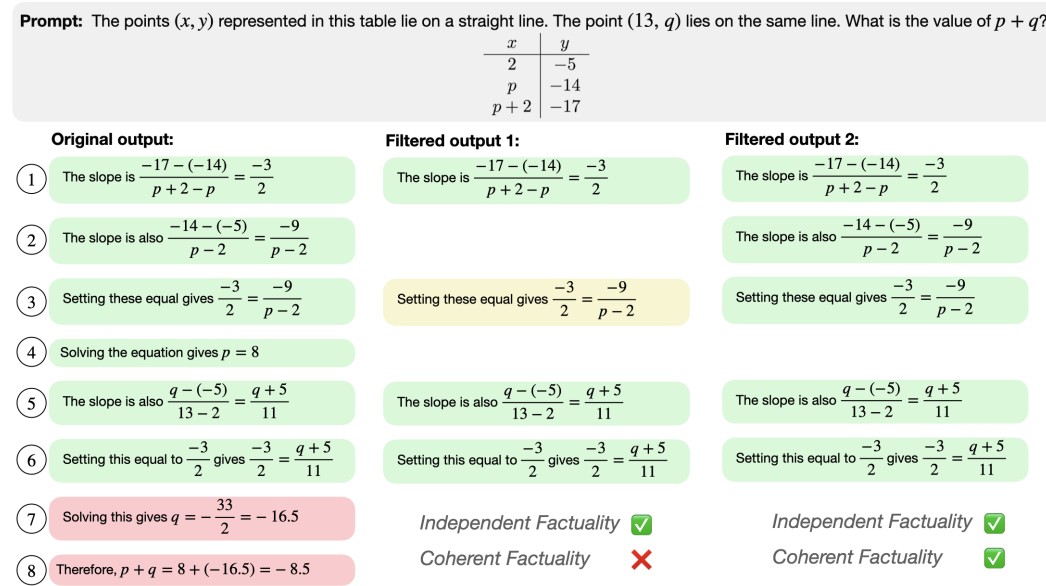

Figure 1: Here, the previous method (Output 1) removes the erroneous claims outlined in red, but leaves the response incoherent by removing Step 2, which is referenced in Step 3. We (Output 2) consider reasoning structure to filter out erroneous claims while maintaining coherence; even though we remove a true claim, it is not essential for understanding the claims that remain ($\alpha = 0.1$).

that language model generations are both *factual* and *coherent* by evaluating entire orderings of claims as correct rather than evaluating individual claims.

**An algorithm for coherent claim filtration.** To apply this *coherent* definition of factuality, we propose a graph representation for inter-claim dependence and an empirical method for obtaining such a graph. Rather than filtering claims individually, we filter between "well-supported" subgraphs via split conformal prediction to ensure coherence and factuality at any user-specified rate.

**Empirical realization of conformal guarantees.** We validate our algorithm on a variety of competition math problems from the MATH dataset (Hendrycks et al., 2021) and from FELM (Chen et al., 2023a), and experiment with different heuristic risk functions. We find that our graphical representation is often both *sufficient* (graph-based calibration satisfies conformal guarantees) and *necessary* (calibration that ignores graph structure does not satisfy conformal guarantees) to ensure coherent factuality. We achieve outputs as complete as the baseline with improved "legibility," or third-party verifiability, and we bootstrap filtered responses by reprompting to further improve factuality.

## 1.1 RELATED WORK

**Conformal prediction** is a statistical uncertainty quantification technique which yields marginal coverage guarantees over a confidence set in a distribution-free manner, traditionally only assuming exchangeability of the data (Gammerman et al., 1998; Shafer & Vovk, 2008; Angelopoulos & Bates, 2022). Split conformal prediction (Papadopoulos et al., 2002; Lei et al., 2018; Romano et al., 2019) is a batched algorithm which relies on a held out calibration set to yield tight guarantees in expectation over the draw of the calibration set. While conformal prediction has been explored under graph settings, this has largely been in the context of hierarchical labels (Tyagi & Guo, 2024; Angelopoulos et al., 2023) or graph neural networks, rather than induced graphs for reasoning.

Recent work has sought to apply conformal prediction to language modeling, including multiple choice question answering (Kumar et al., 2023), as well as open domain and domain-specific question answering and long-form generations (Quach et al., 2024; Mohri & Hashimoto, 2024; Cherian et al., 2024; Liu & Wu, 2024).

Mohri & Hashimoto 2024 applies nested conformal prediction (Gupta et al., 2022) with entailment sets, splitting generations into disjoint claims, and obtaining confidence estimates for each such that removing claims below a corresponding calibrated threshold score yields an $\alpha$-conformal factual

response. Cherian et al. 2024 extends this framework by introducing level adaptivity by conditional calibration (see also Detommaso et al. (2024) for a conditional calibration approach to scoring factuality), lowering the correctness level while simultaneously ensuring completeness of the output. Liu & Wu 2024 extend Mohri & Hashimoto 2024 to give context-conditional coverage guarantees using the group conditional conformal prediction techniques developed by Jung et al. 2023. However, while these works are effective in their application domains, where claims may be treated as independent, they do not generalize to reasoning problems, where the correctness of each step cannot be evaluated without the context of the steps that precede it.

**LLM Reasoning.** Chain-of-Thought (CoT) reasoning induces LLMs to produce step-by-step rationales to support their generations, similar to the human System 2 reasoning process (Wei et al., 2024; Nye et al., 2021; Kojima et al., 2022). Several approaches have been proposed to explore thought diversity to this effect by sampling more and marginalizing over reasoning chains (Wang et al., 2023b; Chen et al., 2023b), incorporating different types of feedback (e.g. self-critique, external verifiers) and revision (Yao et al., 2023; Besta et al., 2024).

(Radhakrishnan et al., 2023) introduced CoT decomposition and factored decomposition as procedures that iteratively solve subquestions that make up the final generation, and showed that while accuracy drops slightly, factored decomposition greatly improves faithfulness to the true reasoning process of the model. Lastly, works on process supervision and intermediate verification (Lightman et al., 2023; Ma et al., 2023; Dhuliawala et al., 2023) help with mitigating hallucination, but are costly at test-time and rely on the correctness of the feedback. We show how our filtered output can be used as chain-of-thought to get more factual completions.

## 2 PRELIMINARIES

**Setup and notation.** As is standard in the language model (LM) generation setting, we assume that the LM takes in input $X \in \mathcal{X}$ and generates an output $Y \in \mathcal{Y}$. We further assume that an output $Y$ can be written as set of "claims," and our goal is to filter the output to keep a set of "factual" and "coherent" claims. Note that we do not attempt formal definitions for each of these difficult terms, and we ultimately evaluate our method's performance with human annotations.

**Definition 1** (Claim). *A claim is an atomic proposition. From this, we define $\mathcal{C}$, the set of all claims.*

For example, claims might assert things like "The sky is blue" or, more abstractly, provide the definition of addition. The set of claims $\mathcal{C}$ can also contain assertions that are incorrect–for example that "Barack Obama was president in 2020." Note that we will not formalize where the boundaries are for what makes a particular string an atomic "claim" or not; we assume access to a *claim splitter function*, which takes LM outputs in $\mathcal{Y}$ and maps them to a set of discrete claims. We write this as $S : \mathcal{Y} \to 2^{\mathcal{C}}$. In practice, we will use a language model to implement claim splitting (Figure 1).

**Definition 2** (Ground truth). *The ground truth $C_{\text{true}} \subseteq \mathcal{C}$ is the subset of all claims we assume to be valid without any additional information or context. In particular, this set is some known body of knowledge from which we base our evaluations of factuality.*

**Remark 1.** *In practice, we might choose some reference like Wikipedia or a math textbook as our ground truth. It is important to note that the ground truth is not necessarily fixed over examples and can be context-sensitive–for instance, while it is generally reasonable to assume that $\sqrt{2}$ is irrational, it is not reasonable to do so in a proof of that fact.*

**Background: Conformal prediction guarantees for LM generations.** Mohri & Hashimoto 2024 improve the factuality of LM generations by splitting them into subclaims and filtering low-confidence subclaims via conformal prediction. They obtain factuality calibrated to a user-specified parameter $\alpha$ while maintaining a significant proportion of the original output. Each subclaim is scored according to some heuristic confidence function[2] $\sigma : \mathcal{C} \to [0, 1]$ computed by comparing particular subclaims to alternate generations for the same prompt. For each output, the non-conformity score $r(X, Y, \mathcal{T})$ is simply the minimum threshold in a set $\mathcal{T}$ such that all subclaims with confidence scores above the threshold are "factual" (or entailed by the ground truth $C_{\text{true}}$, as verified by a human annotator). Further mathematical details are in Appendix H.

---

[2]To frame our filtering method as incurring risk by adding subclaims, we instead consider $\sigma$ to be a heuristic risk function–details follow.

Then, for a calibration set of $(X_1, Y_1), ..., (X_n, Y_n)$, ordering $r(X_1, Y_1, \mathcal{T}), ..., r(X_n, Y_n, \mathcal{T})$ and taking $\hat{q}_\alpha$ as the $\frac{\lceil (n+1)(1-\alpha) \rceil}{n}$ quantile of the scores we obtain the split conformal guarantee:

$$1 - \alpha \leq \mathbb{P}[r(X_{n+1}, Y_{n+1}, \mathcal{T}) \leq \hat{q}_\alpha] \leq 1 - \alpha + 1/(n+1).$$

This result assumes data exchangeability and no ties in scores (which can be enforced by inserting continuous noise). Mohri and Hashimoto further assume that $(\forall y \in S(Y), C_{\text{true}} \implies y) \iff (Y \text{ is factual})$, i.e., the factuality of $Y$ is simply the simultaneous factuality of each of its claims $y$. Then, by omitting claims in $S(Y_{n+1})$ with confidence scores below $\hat{q}_\alpha$ and recombining the remaining claims in a filtered $Y_{n+1}$ which we denote $Y_{n+1}^{\hat{q}_\alpha}$, the above guarantee transfers to factuality.

# 3  A New Notion of Factuality: Coherent Factuality

While the approach of Mohri & Hashimoto 2024 calibrates to a useful notion of factuality, this notion implicitly makes the strong assumption that subclaims are independent, so we call it *independent factuality*. Specifically, the assertion that $(\forall y \in S(Y), C_{\text{true}} \implies y) \iff (Y \text{ is factual})$ treats each claim's correctness independently of the other claims in the generation. While this may be appropriate for pure recall tasks, like biography generation, we find that it is not sufficient to preserve output quality for reasoning tasks. Our notion of coherent factuality further imposes coherence by requiring both correctness *and* substantiation.

**Definition 3** (Coherent factuality). *Given an example $X$ and ground truth $C_{\text{true}}$, an output $Y_{\text{ordered}} = (y_1, ..., y_n) \in \mathcal{C}^{\mathbb{N}}$ of distinct claims is coherently factual if it satisfies*

$$\forall i \in [n], y_i \text{ is deducible from } (y_1, ..., y_{i-1}), X, C_{\text{true}}.$$

We omit a formal definition for "deducible" because deducibility is both subjective and context-sensitive (a claim may follow immediately for professional mathematicians but not for grade-schoolers, unless they are very precocious). Note that we require a claim in the ordering to be deducible from its prefix, the ground truth, *and* the example $X$, since information like variable definitions will be sensitive to the context. As noted before, the ground truth is determined in part by the question (it is not appropriate to assume a fact in the proof of that fact).

**Remark 2.** *According to this definition, coherence cannot come at the cost of factuality. Deducibility is only stricter than implication; in particular, any fact which is deducible from the ground truth must be implied by the ground truth. At worst, we might expect that by calibrating for this more stringent notion, we would simply output subsets of the claims output by the previous method. However, by making use of graphical structure in our scoring and filtering, our method produces outputs of similar completeness to those of Mohri & Hashimoto (2024) and which, in some cases, contain important reasoning steps the previous method had omitted (see Appendix L).*

Like independent factuality, coherent factuality does not stipulate that the response is relevant or responsive to query $X$ (although it cannot contradict $X$), and would therefore consider logically consistent non-sequiturs to be correct. In the setting we consider, we find that requiring relevance is not necessary, since the LMs we study consistently attempt a relevant response.

Intuitively, coherent factuality ensures outputs contain sufficient reasoning between previous claims and subsequent ones and considers *orderings* of claims rather simply sets. Steps must appear in logical sequence. For instance, a variable must be defined before it is used. Given a set of claims $S(Y)$, we write $\pi(S(Y)) \in \mathcal{C}^{\mathbb{N}}$ to denote a particular ordering of those claims.

**Observation 1.** *If an ordering $(y_1, ..., y_n)$ is coherently factual, any prefix $(y_1, ..., y_i)$ for $i < n$ is also coherently factual.*

## 3.1  Graphical Representations of Coherent Factuality

It will be helpful for us to capture coherence graphically. To do so, we will make the following benign assumption: if a claim is deducible from some information, the claim remains deducible after adding more "good" information.

**Assumption 1** (Superstring deducibility). *Fix some input $X$, ground truth $C_{\text{true}}$ and claim $y_n$. Say that $y_n$ is deducible from some ordering of $\{y_1, ..., y_{n-1}\}$, and call the ordering $Y_{\text{sub}}$. Then, if $Y_{\text{super}}$ is a coherently factual ordering on a superset of $\{y_1, ..., y_{n-1}\}$, $y_n$ is also deducible from $Y_{\text{super}}$.*

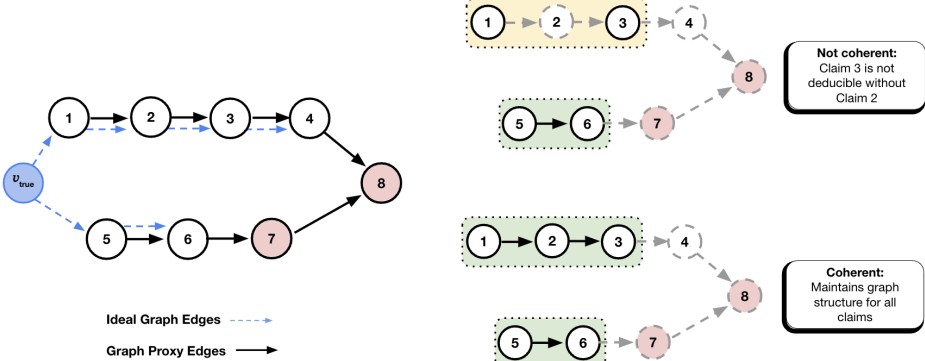

Figure 2: The nodes above correspond to the subclaims enumerated in Figure 1. In blue is the ideal deducibility graph for this output which gives perfect information and allows us to keep all true claims. Even though our approximate deducibility graph lacks a ground truth node and has additional edges (e.g., $(6, 7)$), it helps us preserve the integrity of an output while filtering. In contrast, the baseline method leaves Claim 3 unsubstantiated by omitting Claim 2.

**Ideal deducibility graphs.** For a particular $(X, Y), C_{\text{true}}$, an oracle with perfect understanding of the ground truth could construct an ideal deducibility graph $G = (V, E)$. Define vertex set $V := \{S(Y), v_{\text{true}}\}$, with $v_{\text{true}}$ to stand in for all claims in $C_{\text{true}}$ and question $X$ (as claims may be deducible from either/both of these). Then, edges indicate that a claim is deducible from its ancestors. In particular, the oracle could construct the edge set $E$ by iteratively considering topological layers beginning at the ground truth, asking, "Which claims are deducible from previous layers?" and drawing corresponding edges (a more detailed algorithm for construction is in Appendix A).

**Remark 3.** *There may be multiple ideal deducibility graphs. For example, if a claim $c$ is deducible from $a$ or $b$, both deducible from $v_{\text{true}}$, there is no way to represent this relationship uniquely without a hypergraph; a graph with edge $(a, c)$ or $(b, c)$ could be obtained by the algorithm in Appendix A.*

This idealized construction yields a directed acyclic graph (DAG) where substantiated claims descend from $v_{\text{true}}$, and erroneous or unsubstantiated claims do not. If we had such a graph, conformal filtering would be unnecessary; we would simply output the descendants of $v_{\text{true}}$ in topological order. However, this ideal is unattainable, as ground truth and deducibility are not easily defined. Instead, we develop approximations of these graphs that suffice to achieve coherent factuality.

**Approximate deducibility graphs.** We define a weaker notion of an approximate deducibility graph and find this notion is satisfied by GPT-generated proxies. This weaker notion is sufficient to maintain coherence during filtering while ensuring calibrated guarantees on factuality. Unlike ideal graphs, these proxies do not trace claims to a ground truth or represent the minimal set needed to substantiate a claim; instead, they capture *sufficient* sets for substantiation (see Observation 2 for a formal definition of minimality). While they don't tell us which nodes are erroneous, they indicate which claims are "required" for another and give a natural (topological) ordering on the claims.[3]

**Definition 4** (Approximate deducibility graph). *Let $G = (V, E)$ be a DAG for $(X, Y), C_{true}$. Each node $v \in V$ represents a claim $y \in Y$. The edge set $E$ must satisfy the following: (1) Ancestor-connected subgraphs: for any subgraph $G_{sub} = (V_{sub}, E_{sub})$ that includes all ancestors of its nodes, if a coherently factual ordering exists for $V_{sub}$, then every topological ordering of $G_{sub}$ must also be coherent, and (2) Consistency: if an ancestor-connected subgraph $G_{sub}$ does not allow a coherently factual ordering, then any larger subgraph $G_{super} \supseteq G_{sub}$ must not admit a coherent ordering.*

In other words, we require that a particular claim is sufficiently substantiated by its ancestors (so a topological sort on those nodes will be coherently factual if and only if the set does not contain erroneous claims). Since we assume we can access one such graph for each example, we would like to be assured that a graph satisfying this definition can always be constructed.

---

[3]In practice, these will almost always correspond to the original numeric orderings of the claims as originally generated.

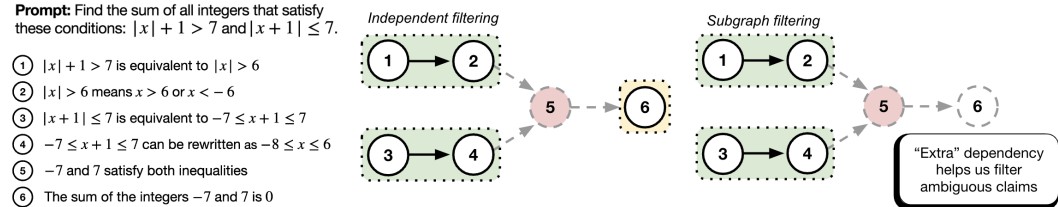

Figure 3: Even though Claim 6 is technically true, it detracts from the coherent solution as it is derived from a false claim (which suggests the solution is 0). Although we do not require dependency, the edge $(5, 6)$ prevents consideration of Claim 6 in the absence of Claim 5. This property improves the quality of the subgraphs we consider.

**Observation 2** (Approximate deducibility graph realizability). *For any $(X, Y)$, $C_{\text{true}}$, there exists a graph with vertex set $S(Y)$ satisfying Definition 4.*

The subgraph of the ideal deducibility graph $G = (V, E)$ induced on $V \setminus v_{\text{true}}$ (omitting the ground truth node) is an approximate deducibility graph (proof deferred to Appendix A).

**Remark 4.** *An ideal deducibility graph is* minimal. *Among all approximate graphs for a particular $(X, Y, C_{\text{true}})$, there exists an ideal graph (minus $v_{\text{true}}$) with the minimum number of edges.[4] Approximate graphs result from removing the ground truth node of an ideal graph and adding edges without introducing cycles (following Assumption 1). Approximate graphs enforce sufficient but not necessary substantiation; even so, they show great empirical utility in Section 5, with quantitative results in Appendix F and a qualitative example in Appendix L.*

While an approximate deducibility graph must exist, we further assume that we can construct one for each $(X, Y)$. In practice, we use GPT-4o to generate these graphs after splitting an output into claims, so we cannot enforce graph validity rigorously. However, our GPT-proxies satisfy Definition 4 in practice, which is sufficient for both calibration bounds to hold. We introduce another property of GPT-generated graphs which we call "dependency." Dependency helps us discard "unreasonable" subgraphs early in our filtering procedure. See Section 5 (end of paragraph "Approximate deducibility graph generation") for more details and Figure 3 (Appendix L) for an example.

## 4 A Protocol for Coherent Factuality

If we had ideal deducibility graphs for each $(X, Y)$, optimal filtering would be easy. Then, we could simply output a topological sort of descendants from the ground truth node and omit the rest. Of course, approximate deducibility graphs don't allow this. They have two essential shortcomings: (1) they may contain extraneous edges (which is preferred over failing to capture dependencies), and (2) they do not identify which claims follow from the ground truth.

**First approach: post-hoc filtering.** We would like to apply conformal prediction to filter the original output while maintaining calibration guarantees. As a first approach, which we call "post-hoc filtering," we take outputs filtered by the independent conformal baseline and apply our graphs to further filter claims lacking their ancestors. This alternate method will achieve coherent factuality by design if our graph proxies are good but may exceed the conformal upper bound as we remove additional erroneous claims.

**Second approach: subgraph filtering.** To achieve calibrated coherent factuality, we compute risk thresholds over a set of subgraphs of the approximate deducibility graph $G$ to consider which subgraph (and corresponding topological ordering of claims) to output. We subsequently show that thresholding based on this set suffices to obtain conformal coherent factuality.

To select subgraphs, we use a heuristic risk-scoring function $\sigma : \mathcal{C} \rightarrow [0, 1]$, which differs from Mohri & Hashimoto (2024) by measuring risk rather than confidence and using the graph $G$ as input (elided for simplicity) rather than a singular subclaim. Subgraphs are generated by thresholding

---

[4]Since ideal graphs for the same $(X, Y), C_{\text{true}}$ may have different numbers of edges, it is not the case that *every* ideal graph is minimal in this way.

---

**Algorithm 1:** Subgraph Generator

---

**Input:** Graph $G = (V, E)$, claim-wise risk function $\sigma : V \to \mathbb{R}$
**Output:** $\mathcal{U}_\mathcal{T} :=$ set of subgraph, threshold pairs $(U_i, \tau_i)$
$\mathcal{U}_\mathcal{T} \leftarrow \emptyset$, $\mathcal{T} \leftarrow$ sorted($\{-\infty\} \cup \{\sigma(v) \mid v \in V\}$) // `Sort risk scores`
**foreach** $\tau_i \in \mathcal{T}$ **do**
    $V_i \leftarrow \{v \in V \mid \sigma(v) \le \tau_i\}$ // `Select nodes below threshold`
    **foreach** $v \in V_i$ *in topological order* **do**
        **if** $\exists$ *ancestor of $v$ not in $V_i$* **then**
            $V_i \leftarrow V_i \setminus \{v\}$ // `Remove claim with missing ancestors`
    $U_i \leftarrow G[V_i]$ // `Induced subgraph`
    $\mathcal{U}_\mathcal{T} \leftarrow \mathcal{U}_\mathcal{T} \cup \{(U_i, \tau_i)\}$
**return** $\mathcal{U}_\mathcal{T}$

---

nodes independently and filtering out vertices lacking ancestors, producing at most $|S(Y)| + 1$ induced subgraphs. The heuristic risk of each subgraph corresponds to its threshold, with at most $n + 1$ relevant thresholds, one for each each node and one for the empty set (Algorithm 1).

**Scoring functions.** Claim retention depends on our choice of claim-scoring function $\sigma$. We apply a context-independent claim-scoring function $\sigma_{\text{ind}}$ to score nodes individually. We refer to $\sigma_{\text{ind}}$ as *self-consistency* scoring. In practice, we compute $\sigma_{\text{ind}}$ as in Mohri & Hashimoto (2024) by querying GPT-4 to generate 5 alternate responses and counting the frequency with which each subclaim appears (prompt in K.1). We flip these confidence scores to obtain risk scores and use node risk scores to compute $\sigma$ in the following two[5] ways (with the use of the graph $G$):
(1) *Graph independent:* $\sigma(v) = \sigma_{\text{ind}}(v)$, which does not consider the graph to score each node.
(2) *Descendant weighting:* For each $v \in V$, define $\sigma(v) = (1 - \beta)\sigma_{\text{ind}}(v) + \beta\,\text{median}\{\sigma_{\text{ind}}(v') : v' \text{ is a descendant of } v\}$, where $\beta$ is a hyperparameter[6]. The motivation for the descendant weighting function is to boost (reduce) confidence if the claims derived from a particular claim are very confident (uncertain).

Once we have a set of subgraphs $\mathcal{U}$ corresponding to an output $Y$, the non-conformity score of $Y$ is simply the risk threshold below which all subgraphs make "good" filtered outputs.

**Definition 5** (Non-conformity scoring function). *Given some $(X, Y)$ pair, deducibility graph $G = (V, E)$, candidate subgraphs and thresholds $\mathcal{U}_\mathcal{T} \subseteq \mathcal{U} \times \mathcal{T}$, we compute non-conformity score as follows:*

$$r(X, Y, \mathcal{U}_\mathcal{T}) = \sup\{\tau_r \in \mathbb{R} \mid \forall (U, \tau) \in \mathcal{U}_\mathcal{T} \text{ with } \tau \le \tau_r, U \text{ is coherently factual}\}$$

In other words, $r(\cdot)$ is the maximum tolerable risk: the risk of the first subgraph violating coherent factuality if one exists, otherwise $\infty$. Also, "$U$ is coherently factual" is shorthand for "each topological sort of $U$ is coherently factual according to $X, C_{\text{true}}$."

**Conformal correctness guarantees.** Now, to apply split conformal prediction to control this risk, we take $\hat{q}_\alpha := \frac{\lceil (1-\alpha)(n+1) \rceil}{n}$th quantile of $\{1 - r(X_i, Y_i, \mathcal{U}_{\mathcal{T}_i})\}_{i=1}^n$. We then filter new outputs $(X_{n+1}, Y_{n+1})$ with $G_{n+1}$ by generating $\mathcal{U}_{\mathcal{T}_{n+1}}$, computing

$$U_{\text{filtered}}, \tau_{\text{filtered}} = \underset{(U, \tau) \in \mathcal{U}_{\mathcal{T}_{n+1}} \mid \tau < 1 - \hat{q}_\alpha}{\arg\max} \tau,$$

and defining our final filtered output $Y_{n+1}^{\hat{q}_\alpha} := V'_{\text{filtered}}$, a topological sort on $V_{\text{filtered}}$.[7] With the minimal assumption of exchangeability of the underlying distribution $\mathcal{D} = \mathcal{X} \times \mathcal{Y}$, we have the following theorem (see Appendix C for full proof).

---

[5]Note that there are several other ways to use graph structure for scoring (including modifications of the ones below). We leave further exploration to future work.

[6]We explored several similar graph-sensitive scoring mechanisms, each motivated by weighting the risk score of a node according to the risk scores of its ancestors and/or descendants. The median version was most robust in performance to small changes in beta (we speculate this is because the median is not sensitive to outlier scores). We swept beta values in [0, 1] and chose 0.5 for its good performance.

[7]If there are no back edges ($y_j \rightsquigarrow y_i$ when $j > i$) in the "original" ordering of claims, removing the filtered claims without altering ordering yields a valid $V'_{\text{filtered}}$.

**Theorem 1** (Calibrated Factuality). *Fix some calibration set $\{(X_i, Y_i)\}_{i=1}^n$, test point $(X_{n+1}, Y_{n+1}) \sim \mathcal{D}$, ground truth $C_{\text{true}}$, and desired error rate $\alpha$. Then the following holds:*

$$1 - \alpha \leq \mathbb{P}[Y_{n+1}^{\hat{q}_\alpha} \text{is coherently factual}].$$

*If, additionally, each $G_i$ is an approximate deducibility graph (see Definition 4) and $r(X, Y, \cdot) < \infty \; \forall (X, Y)$ [8], we have:*

$$\mathbb{P}[Y_{n+1}^{\hat{q}_\alpha} \text{is coherently factual}] \leq 1 - \alpha + \frac{1}{n+1}.$$

## 5 EMPIRICAL FINDINGS

**Datasets.** Our experiments make use of the MATH dataset (Hendrycks et al., 2021), which spans various branches of mathematics. This dataset is among the standard benchmarks reported in recent model releases, on which even frontier models hallucinate. We also use the FELM dataset Chen et al., 2023a which consists of a variety of verbal reasoning problems with results in Appendix D. We replicate our main experiments with an open-source model (Llama-3.1-70B-Instruct for output and graph generation) and discuss the costs associated with GPT prompts in J.

**Approximate deducibility graph generation.** For proprietary models, we used examples and outputs from Mohri & Hashimoto (2024), where subclaims were generated by GPT-4. We then queried GPT-4o via few-shot prompting (Appendix K) to produce adjacency lists, as graph generation proved more challenging than claim-splitting. Open-source experiments followed a similar setup (Appendix K.1). Dependency is an empirically useful heuristic suggesting the consideration or use of one claim in producing the other, whether or not the use was *correct*. In this way, a claim might depend on another even if it results from a logical misstep. Dependency structure is quite common among the subgraphs we generate: in fact, $50\%$ of graphs that contain any erroneous nodes have edges between erroneous nodes. For evidence of dependency's empirical utility, see Appendix F for quantitative data and L for qualitative data.

**Annotation.** *Individual claim* (silver standard) and *subset-level* (gold standard) annotations were used to evaluate output factuality. For individual claims, annotators assessed whether a claim $c$ would be true if all its graph ancestors were true, or, for *a priori* claims, whether it was supported by the ground truth. Subset factuality was measured by checking (1) ancestor connectedness and (2) whether any claim in the subset had an individual annotation of "No," assuming the graph proxies are reliable—an assumption that may falter with sparse representations. Gold standard annotations directly assessed subsets for human notions of coherent factuality, independent of the graph. Silver annotations demonstrate the utility and accuracy of deducibility graphs through relative calibration. The MATH dataset includes both annotation types, while FELM includes only silver annotations.

**Results.** We directly compare the results of our coherent calibration algorithm with the conformal factuality algorithm of (Mohri & Hashimoto, 2024), which we call the *baseline*, on both independent and coherent definitions of factuality, considering the samples from the MATH dataset as well as the FELM dataset. We validate all of our methods on manual (gold standard) annotations on each output. We also test our methods on the FELM Dataset [9] with results in D and demonstrate utility for varying types of reasoning problems. We attempted to generate deducibility graphs for the FActScore biography-generation dataset; however, we found these graphs to be nonsensical and to contain cycles as responses to such prompts do not carry any inherent, directed structure. Our prompts can be found in Appendix K.1, and results of these experiments with Llama-3.1-70B-Instruct can be found in Appendix J.

**[R1] Graph proxies are *sufficient* to obtain coherent factuality.** The quality of the graph proxies is affirmed by the empirical satisfaction of theoretical guarantees in Figure 4a. Both bounds hold across factuality levels when we calibrate on silver annotations that assume proxies are good and validate on gold annotations. Empirical measurements of graph quality are in F. . We note some miscalibration

---

[8]This means each output contains a hallucination. This assumption is implicit in Mohri & Hashimoto (2024) and describes the setting in which this technique is most useful. The lower bound holds without this assumption, and the upper bound approximately holds if the underlying hallucination rate $> \alpha$ (we are working on a generalization of this upper bound).

[9]This dataset contains reasoning word problems.

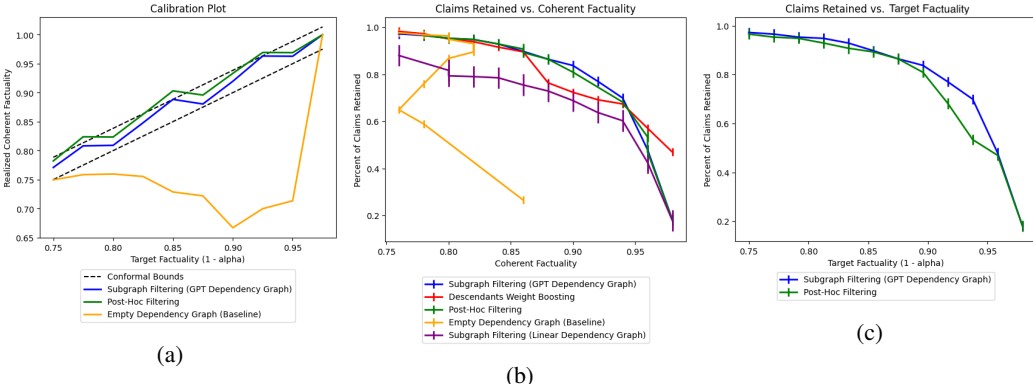

Figure 4: We evaluate our post-hoc (green) and subgraph filtering algorithms (using descendant weighting with $\beta = 1/2$ (red) and graph-independent scoring (blue)) on MATH dataset. Post-hoc filtering is applied using the graph after initial filtering without a graph. We consider the baseline to be the method of Mohri & Hashimoto (2024) (yellow). In (a), we show calibration to desired factuality levels for Subgraph Filtering within theoretical bounds (shown in grey). In (b), we assess claim retention rates by varying $\alpha$ values, plotting both realized factuality and the fraction of retained claims across calibration methods and graph generation techniques. In (c) we plot claim retention with respect to user-desired calibration level.

for the FELM dataset (see Appendix 5a), which could be due to the lack of gold standard annotations for validation or incorrect graphs. The more efficient annotation method contingent on good LM-generated graphs gives a practical empirical instantiation of our algorithm.

**[R2] Graphical proxies are *necessary* to obtain coherent factuality.** The baseline method fails to achieve both calibration (Figure 4a) and competitive claim retention likely because independent factuality does not often imply factual coherence. However, we must still validate against a simple deducibility graph $1 \to 2 \to \cdots \to N$ following the (linear) order in which claims occur in the generation, which also fails to achieve competitive levels of claim retention for the majority of $\alpha$ values when compared to subgraph filtering. The linear method performs better on the FELM dataset (Figure 5b), which suggests the underlying graphs are closer to linear than they are in MATH.

**[R3] Post-hoc filtering is not calibrated.** While post-hoc filtering achieves similar claim retention as subgraph filtering for a *realized* factuality level, it is not calibrated to user input. For a fixed *user-specified* factuality rate (which post-hoc filtering will often overshoot), subgraph filtering achieves better claim retention than post-hoc filtering although post-hoc filtering shows potential to correct independently-calibrated outputs. We note similar lack of calibration in post-hoc filtering for the FELM dataset (Figure 5a).

**[R4] Conformally-filtered results achieve high levels of factuality while retaining most claims.** We empirically achieve high coverage levels while retaining a majority of claims, thus preserving the utility of the generation (Figure 4b). This is important as conformal guarantees can trivially achieved by removing all claims with some calibrated probability. For example, the subgraph filtering algorithm obtains 90% factuality while retaining close to 80% of the claims, and obtains 85% factuality while retaining nearly 90% of the claims. The descendant weighting scoring function shows superior performance at low $\alpha$, achieving arbitrarily high factuality while retaining at least 40% of claims.

**[R5] Coherent outputs are more "legible" than the baseline while equally complete.** (Kirchner et al., 2024) define legible reasoning as "reasoning that is clear and easy to check." We defer human studies of output legibility to future works, but as a proxy, we asked GPT-4o and Llama-3.1-70B-Instruct to grade filtered outputs as either correct or erroneous (more details in Appendix N). For each combination of output generation model (GPT-4, Llama-3.1-70B-Instruct) and output grading model (aforementioned judges), our method was more legible than the baseline (lower false positive and false negative rates for fixed levels of factuality). This improved output utility does not come at the cost of completeness: at $\alpha = 0.1$, **64%** of error-free outputs contain a correct final answer, the *same rate* as the baseline outputs, which have diminished legibility and coherence.

Table 1: Change in error rate on questions with reprompting using claims retained via coherent and independent methods. We record the error rate of GPT outputs on the prompt before conformal prediction is applied (zero-shot) and the error rate of GPT outputs when prompted to complete an incomplete (filtered) output. We compare error reduction between coherent incomplete outputs and incoherent incomplete outputs.

| | Coherent Factuality Error | | | Independent Factuality Error | | |
|---|---|---|---|---|---|---|
| $\alpha$ | *Zero-shot* | *Post-filter* | *Reduction* | *Zero-shot* | *Post-filter* | *Reduction* |
| **0.05** | 28% | 10% | ↓18% | 28% | 26% | ↓2% |
| **0.10** | 28% | 10.88% | ↓17.12% | 28% | 16.56% | ↓11.44% |
| **0.15** | 28% | 14% | ↓14% | 28% | 18.84% | ↓9.16% |

**[R6] Bootstrapping coherently factual inputs improves factuality of regenerations.** We bootstrap coherent factuality by running the filtered output back through the model with the original prompt and requesting the model to fill in the blanks of our filtered output. See I for more details. For $\alpha = 0.05, 0.10, 0.15$, reprompting on coherent outputs provides consistently better reductions in error rate, as compared to independently filtered outputs (Table 1). We posit this methodology is more effective for coherent outputs because they are easier to parse and build upon, demonstrating the utility of our method.

## 6 DISCUSSION

We show how to achieve coherent factuality using the underlying graph structure of deducibility in reasoning problems. We show both theoretical bounds on the calibration guarantees of our method, and practical utility of our approach to improve factuality of language models. Here we discuss limitations and potential future directions.

**Graph proxies.** While our graph proxies satisfy the definition of deducibility graphs empirically, relying on a proprietary model like GPT-4o for accurate graph generation is not ideal. We note that GPT-4o struggled with longer reasoning outputs containing many claims, raising concerns about practicality for multi-step problems.

**Subjective ground truth and deduction.** Whether a claim is valid depends on the annotator's perspective and context. In a complex theorem, arithmetic may be implicit, while for simple algebra, it could be central. Assumptions and axioms also vary by context. It is important to note that correctness of outputs is only consistent with the annotator's subjective notion of truth.

**Improved scoring functions.** Our method works with any subgraph scoring function and increases claim retention by working to converge on the "true" underlying risk function with our scoring function. Improvements may include scoring subsets beyond those considered by our algorithm based and accounting for additional graph structure in node heuristic measures.

**Expanding evaluation to further domains.** This work is primed to extend to any reasoning context, where a graphical representation is not insignificantly sparse. For example, code generation is a natural domain, as compilation is both an easy and well defined notion of coherent substantiation, and correct final outputs clearly indicate correctness. Furthermore, dependency graphs are a common notion in software systems at large, which pairs well with our framework.

## ACKNOWLEDGMENTS

SG acknowledges funding from OpenAI SuperAlignment Fast Grant as well as Microsoft Research grant which generously supported the compute resources required in this work.

MG acknowledges funding from Penn Engineering's Rachleff Scholars Program.

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

## A  GRAPH DETAILS

---

**Algorithm 2:** Ideal Graph Assembly

---

**Input:** example $(X, Y)$, ground truth $C_{\text{true}}$, claim splitter $S : \mathcal{Y} \to 2^{\mathcal{C}}$

$V_{\text{start}} = S(Y)$
$V, E = \{\}$
$L_0 = \{v_{\text{true}}\}$
$V = V \cup L_0$
$t = 0$
**while** $\exists v \in V_{\text{start}}$ that is deducible from some ordering of nodes in $V$ **do**
 $\quad t \leftarrow t + 1$
 $\quad L_t = \{\}$
 $\quad$ **for** each such $v$ **do**
 $\quad\quad$ ancestors $= \{A \subseteq V \,|\, \exists \pi$ with $\pi(A), v$ is coherently factual$\}$
 $\quad\quad A = \arg\min_{A' \in \text{ancestors}} |A' \cap L_{t-1}|$
 $\quad\quad L_t = L_t \cup v$
 $\quad\quad V_{\text{start}} = V_{\text{start}} \setminus \{v\}$
 $\quad\quad$ **for** each $v' \in A \cap L_{t-1}$ **do**
 $\quad\quad\quad E = E \cup \{(v', v)\}$
 $\quad V = V \cup L_t$
**return** $G = (V, E)$

---

*Proof of approximate deducibility graph existence.* To prove this, we first make note of an important property of the ideal graph construction.

**Lemma 1.** *In an ancestor-connected subgraph of the ideal graph, a claim $v$ is a descendant of $v_{\text{true}}$ iff. it is an element of a coherently factual ordering.*

Assume a node $v$ is in the vertex set $V_{\text{sub}}$ of ancestor-connected subgraph $G_{\text{sub}} = (V_{\text{sub}}, E_{\text{sub}})$ of $G_{\text{ideal}}$.

To prove the forward direction, assume $v$ is a descendant of $v_{\text{true}}$ in $G_{\text{sub}}$. Then, by construction, there is some ordering $\pi(a(v)) = (v_{\text{true}}, v_1, ..., v_k)$ such that $(v_{\text{true}}, v_1, ..., v_k, v)$ is coherently factual.

For the backward direction, assume that $v$ is part of a coherently factual ordering $(v_1, ..., v_k, v)$. Then, by the definition of coherent factuality, $v_1$ is deducible from $v_{\text{true}}$, and so on inductively, so each node preceding $v$ will have a path from $v_{\text{true}}$ in the ideal graph. Thus, $v$ will also be a descendant of $v_{\text{true}}$ in $G_{\text{ideal}}$; since $G_{\text{sub}}$ is ancestor connected, this holds in $G_{\text{sub}}$.

Now, we proceed with the proof of approximate deducibility graph existence. Fix some $(X, Y), C_{\text{true}}, S : \mathcal{Y} \to 2^{\mathcal{C}}$.

Generate the ideal graph $G_{\text{ideal}} = (V_{\text{ideal}}, E_{\text{ideal}})$ with Algorithm 2. Then, consider its subgraph on $V := V_{\text{ideal}} \setminus L_0 = L_1 \cup ... \cup L_n$. Call this $G = (V, E)$.

$G$ is a DAG by construction, so to prove the approximate deducibility property, we fix some $G_{\text{sub}}$ satisfying ancestor connectedness.

To show (1), assume that a coherently factual ordering exists for $V_{\text{sub}}$. By Lemma 1, this implies that each $v \in V_{\text{sub}}$ is a descendant of $v_{\text{true}}$ in $G_{\text{ideal}}$. Then, by construction, each $v \in V_{\text{sub}}$ is deducible from $a(v)$. In a topological sort, $v$ is preceded by each $v' \in a(v)$, and so any topological sort gives a coherently factual ordering.

To show (2), consider the case that $G_{\text{sub}}$ does not allow a coherently factual ordering. Since $v_{\text{true}}$ is in a coherently factual ordering and $G_{\text{sub}} \cup \{v_{\text{true}}\}$ is an ancestor-connected subgraph in $G_{\text{ideal}}$, Lemma 1 says that at least one element $v_{\text{bad}}$ is not contained in any coherently factual ordering (otherwise, each $v$ would be a descendant of $v_{\text{true}}$; by the previous argument, a topological sort of $V_{\text{sub}}$ is then a coherently factual ordering). In particular, a superset of $V_{\text{sub}}$ contains $v_{\text{bad}}$ and therefore has no coherently factual ordering.

Both properties hold, so the proof is concluded.

$\square$

## B  CONFORMAL FILTERING ALGORITHM

In the algorithm below, we refer to Algorithm 1, "Subgraph Generator," simply as "subG."

---

**Algorithm 3:** Coherent Calibration

---

**Input:** Confidence $\alpha$, calibration data $\{(X_i, Y_i)\}_{i=1}^{n}$, output graphs $\{G_i = (V_i, E_i)\}_{i=1}^{n}$

$\tau = \{\}$

**for** $i$ *in* $[n]$ **do**

   $\mathcal{U}_{\mathcal{T}_i} = \text{subG}(G_i)$

   $\tau = \tau \cup \{r(X_i, Y_i, \mathcal{U}_{\mathcal{T}_i})\}$

$\hat{q}_\alpha = \frac{\lceil(n+1)(1-\alpha)\rceil}{n}$ th quantile of $\{1 - \tau_i | \tau_i \in \tau\}$

**return** $\hat{q}_\alpha$

---

## C  PROOF OF THEOREM 1

*Proof.* To show the following, we refer to the notion of ancestor connectedness introduced in Definition 4. Recall that to obtain the upper bound, we assume that $r(X, Y \cdot) < \infty \; \forall (X, Y)$.

Note that, if we apply Algorithm 1 to $G_{n+1}$, each subgraph in output $\mathcal{U}_{\mathcal{T}_{n+1}}$ satisfies ancestor connectedness.

As we proceed, for ease of notation, we simply write $r(X_{n+1})$ for $r(X_{n+1}, Y_{n+1}, \mathcal{U}_{\mathcal{T}_{n+1}})$.

Now, since $(1 - r(X_{n+1}) \leq \hat{q}_\alpha) \iff (r(X_{n+1}) \geq 1 - \hat{q}_\alpha)$, we have

$$1 - \alpha \leq \mathbb{P}[r(X_{n+1}) \geq 1 - \hat{q}_\alpha] \leq 1 - \alpha + \frac{1}{n+1}$$

as a standard split conformal result (where the probability is taken over the draw of the calibration set and $(X_{n+1}, Y_{n+1})$).

To prove the claim, it suffices to show $r(X_{n+1}) \geq 1 - \hat{q}_\alpha \iff Y_{n+1}^{\hat{q}_\alpha}$ is coherently factual.

For both directions, we will consider $U_{\text{filtered}}$ as in Section 4.

For the forward direction, assume $r(X_{n+1}) \geq 1 - \hat{q}_\alpha$. Then, by definition, conformally filtered $Y_{n+1}^{\hat{q}_\alpha}$ is coherently factual (since $r(X_{n+1})$ is defined such that each subgraph with less risk is coherently factual, and $U_{\text{filtered}}$ satisfies this since $\tau_{\text{filtered}} < 1 - \hat{q}_\alpha \leq r(X_{n+1})$). Note that we make no assumptions on the quality of deducibility graphs to obtain this result.

For the reverse direction, we will show the contrapositive. Assume $r(X_{n+1}) < 1 - \hat{q}_\alpha$. Since $r(X, Y, \cdot) < \infty$, there exists a subgraph, threshold $(U_{\text{bad}}, \tau_{\text{bad}})$ with $U_{\text{bad}} = (V_{\text{bad}}, E_{\text{bad}}) \in \mathcal{U}_{n+1}$, $\tau_{\text{bad}} < 1 - \hat{q}_\alpha$; otherwise, the first bad graph would have risk at least $1 - \hat{q}_\alpha$, so the supremum of safe scores $r(X_{n+1})$ would be at least $1 - \hat{q}_\alpha$.

Say $Y_{n+1}^{\hat{q}_\alpha}$ is the vertex set from $U_{\text{filtered}}$. Note that $\tau_{\text{filtered}} \geq \tau_{\text{bad}}$ (since $\tau_{\text{filtered}}$ is the maximum of risks below $1 - \hat{q}_\alpha$).

If $U_{\text{filtered}} = U_{\text{bad}}$, the desired result ($Y_{n+1}^{\hat{q}_\alpha}$ is not coherently factual) follows.

Otherwise, $U_{\text{bad}}$ is a subgraph of $U_{\text{filtered}}$, and both are ancestor-connected, properties obtained by Algorithm 1. In particular, this means $V_{\text{filtered}}$ is a superset of $V_{\text{bad}}$.

Note that $G_{n+1}$ is an approximate deducibility graph and $U_{\text{bad}}$ is an ancestor-connected subgraph with no coherently factual ordering (if it had one, $V_{\text{bad}}'$ in particular would be coherently factual by Definition 4). Additionally, any superset of $V_{\text{bad}}$ has no coherently factual ordering, also by Definition 4. However, $Y_{n+1}^{\hat{q}_\alpha}$ is one such ordering on superset $V_{\text{filtered}}$, which concludes the contrapositive of the backward direction.

We have thus shown that $(r(X_{n+1}) \geq 1 - \hat{q}_\alpha) \iff (Y_{n+1}^{\hat{q}_\alpha}$ is factual), which proves the claim. $\square$

# D   RESULTS FOR FELM DATASET

We present the results of our algorithms on the FELM Dataset, as discussed in the results section. The lines graphed correspond to the same evaluation settings as with the MATH dataset. We note that the Post-Hoc Filtering algorithm remains un-calibrated (even more so) in these results as compared to subgraph filtering which is (almost perfectly) calibrated (see 5a). The slight discrepancy may be due to erroneous graphs as we lack manual annotations. The no dependency baseline performs better in this case, but still fails to meet the lower bound for any value of $\alpha$. The validation results (see 5b also appear to be similar to that of the MATH dataset. However, we now note better performance of linear graphs, implying that reasoning paths may be closer to perfectly linear in this dataset. Post-Hoc and subgraph filtering remain relatively the same, and are still competitive relative to one another in claim retention.

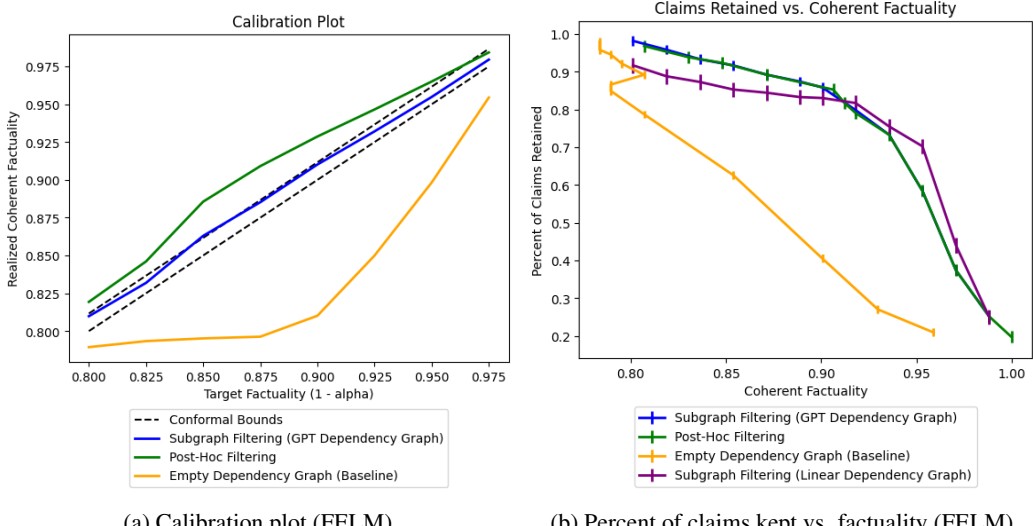

(a) Calibration plot (FELM)          (b) Percent of claims kept vs. factuality (FELM)

Figure 5: Results on the FELM dataset using GPT-4 for responses and GPT-4o for graphs.

# E   RESULTS WITH LLAMA-3.1-70B-INSTRUCT (OPEN-SOURCE)

We ran the same experiment on the MATH dataset for outputs, subclaim splits, and graphs produced by Llama. While Llama-generated graphs were further from ideal and less often satisfied Definition 4 (discussion of our altered approach in Appendix K.1), our empirical results suggest they are still useful. The plots below are for silver-annotated calibration and validation.

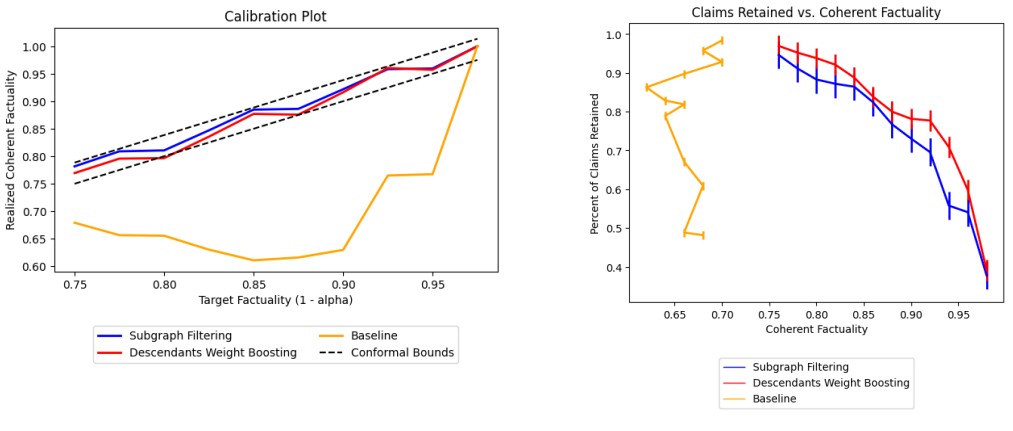

(a) Calibration plot (Llama)          (b) Percent of claims kept vs. factuality (Llama)

Figure 6: Results on the MATH dataset solely using Llama-3.1-70B-Instruct.

## F    MODEL-GENERATED VS. IDEAL GRAPHS

Surprisingly, graphs generated by GPT-4o seemed to have *more* empirical utility than ideal, human-generated graphs. As discussed in Section 5, this additional structure, although not strictly necessary to obtain theoretical guarantees, tends to improve the set of subgraphs we search over by deferring admittance of faulty claims that rely on faulty claims. GPT includes edges between such claims while the ideal construction does not require this.

In Figure 7, we compare the claim retention of GPT-graph vs. ideal-graph calibration on 10 examples from the MATH dataset.

To validate our graphs, for the first 10 GPT proxies, we measure their edit-distance to (manually-constructed) ideal graphs (1.8 on average) and check whether they are approximate deducibility graphs (100% are). The second result means each graph we checked satisfied Definition 4, which is sufficient to obtain both conformal bounds. For the first 10 LLaMa-generated graphs, the edit distance is 10.7, and 40% are approximate deducibility graphs.

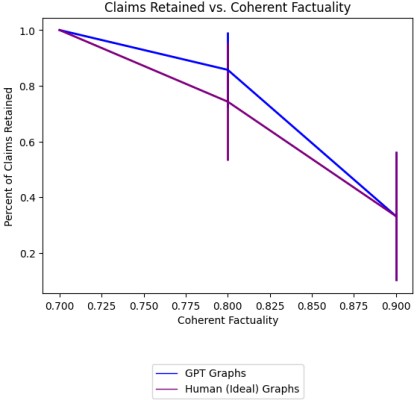

Figure 7: Performance of GPT-generated vs. human-constructed graphs for $\alpha = 0.1, 0.2, 0.3$.

## G    FURTHER RELATED WORK

**Factuality and Hallucination in Language Models.**

Ensuring the factuality of language model outputs is an important objective towards their reliable real-world deployment across diverse settings. Hallucinations can arise in several ways, including lack of knowledge or recall problems from pre-training data, fine-tuning data, or a vector datastore with RAG, as well as issues with decoding strategy (Huang et al., 2023), (Wang et al., 2023a)). Works such as (Kalai & Vempala, 2024) suggest that LMs will always hallucinate while there exists unknown knowledge, while others such as (Ahdritz et al., 2024) seek to identify uncertainty due to lack of knowledge via linear probes. At the same time, (Zhang et al., 2023) demonstrate LLMs can independently identify hallucinations, but often continue with incorrect lines of reasoning even when a mistake is made early on. Our work directly addresses such a setting through dependence-based factuality within a reasoning chain, avoiding cascading hallucinations by design with high probability.

**Uncertainty Estimation.** The problem of insufficient (or incorrect) knowledge can be treated as epistemic uncertainty, while inference-time decoding randomness in sampling can be addressed as aleatoric (Ahdritz et al., 2024). Thus, the study of uncertainty estimation in language models is complementary to our goal of mitigating hallucinations. Prior works have explored expressions of uncertainty including logit weights (Guo et al., 2017; Jiang et al., 2021), surrogate estimates (Shrivastava et al., 2023), sampling variance (Kuhn et al., 2023; Xiong et al., 2024), and natural language generations indicating uncertainty (Lin et al., 2022; Zhou et al., 2023).

There is also a line of work which leverages confidence scores which, when calibrated, should be proportional to the correctness of the generation (Mielke et al., 2022)). Chen & Mueller 2023 use self-reflection and consistency over generations sampled with a fixed temperature, and select the

generation with the highest confidence score (which is also output to the user). Tian et al. 2023 demonstrates that verbalized confidence scores, akin to (Lin et al., 2022), are better calibrated than using the log probabilities, which are generally overconfident relative to the true level of correctness. Band et al. 2024 introduces a pipeline for linguistic calibration with supervised fine-tuning to enable elicitation of faithful confidence scores, and decision-based reinforcement learning through a forecasting formulation. Detommaso et al. 2024 uses multicalibration to several groups of prompt/completion pairs as a means to elicit reliable confidence scores. Our work makes use of a risk function based on our coherent approach on factuality, calibrating with respect to annotated claim and subset labels.

## H    MORE DETAILS ON CONFORMAL FACTUALITY

We expand on the details of Mohri & Hashimoto (2024) application of conformal prediction to language model outputs.

More formally, Mohri & Hashimoto (2024) frame factuality in terms of entailment by the ground truth.

**Definition 6** (Entailment operator). *The function $E : \mathcal{C} \to \{C_{\text{support}} \subseteq 2^{\mathcal{C}}\}$ takes in a claim $c \in \mathcal{C}$ and outputs each set $C_{\text{support}} \subseteq 2^{\mathcal{C}}$ of claims whose conjunction implies $c$.*

*If $C \in E(c)$, we abuse notation and simply write $C \implies c$. Mohri and Hashimoto seek to retain claims $c$ such that $C_{\text{true}} \implies c$ for each $c$, and consider this sufficient for realizing factuality of an output.*

There is some difference in notation between this definition and the original since Mohri and Hashimoto frame the ground truth $C_{\text{true}}$ as simply an element of $\mathcal{Y}$, while we frame it as a set of claims.

**Definition 7** (Independent non-conformity scoring function). *For a particular output $Y$ with claims $C = S(Y)$ and some set $\mathcal{T}$ of candidate thresholds, the non-conformity score $r$ is defined as follows:*

$$r(X, Y, \mathcal{T}) = \inf\{\tau \in \mathcal{T} : \forall j \geq \tau, \forall y \in C, (\sigma(y) \geq j) \implies (C_{\text{true}} \implies y)\}$$

Then, since increasing the threshold can only remove claims, the traditional conformal guarantee

$$1 - \alpha \leq \mathbb{P}[r(X_{n+1}, Y_{n+1}, \mathcal{T}) \leq \hat{q}_\alpha] \leq 1 - \alpha + \frac{1}{n+1}.$$

can be written as

$$1 - \alpha \leq \mathbb{P}[\forall y \in S(Y_{n+1}^{\hat{q}_\alpha}), C_{\text{true}} \implies y] \leq 1 - \alpha + \frac{1}{n+1}.$$

Then, they assume that $(\forall y \in S(Y), C_{\text{true}} \implies y) \iff (Y \text{ is factual})$, so we obtain

$$1 - \alpha \leq \mathbb{P}[Y_{n+1}^{\hat{q}_\alpha} \text{ is factual}] \leq 1 - \alpha + \frac{1}{n+1}.$$

## I    MORE DETAILS ON BOOTSTRAPPING CONFORMAL FACTUALITY

We may use the outputs of both the baseline independent factuality conformal prediction algorithm and our new coherent factuality conformal prediction algorithm to reprompt the model, see K.3 for the exact prompt. We give the model both the original question and the remaining filtered output and ask it to complete the solution using the context given.

For $\alpha = 0.05, 0.1, 0.15$ we observe the change in factuality from prompting with no context to prompting with the added context of the filtered set of claims. After reprompting, we observe the correctness of the new output and record the new error rate. The new output is considered correct only if all the new claims and reasoning used are correct. The table 1 demonstrates how the error rate has a greater reduction when reprompting with a coherent subset of the original claims rather than an incoherent subset.

## J   COSTS ASSOCIATED WITH GPT QUERIES AND RUNNING ON LLAMA-3.1-70B-INSTRUCT

**Cost and reproducibility.** We replicated our main experiments with Llama-3.1-70B-Instruct (for output and graph generation) with slight changes to the prompting required to elicit useful graphs (see Appendix K.1). We find that the utility of the approach holds for less powerful open-source models: we present our results in Appendix .

The algorithm is also inexpensive to implement. For each example in the calibration and test set, the algorithm requires 8 queries comprising at most 16k tokens; for our calibration set of 50 examples, this cost less than \$5.00 using GPT and less than \$0.70 using Llama. The same queries are made for the test set, so each test example cost less than \$0.10 for GPT and \$0.01 for Llama. These estimates are conservative, assuming full utilization of 2000-token total context and output to accommodate longer form responses (although our responses were much shorter). Perhaps more prohibitive than monetary cost is the number of annotations necessary (at worst exponential in $n$, the number of subclaims for an example). However, this is a one-time cost for calibration, and our results suggest that silver annotations, of which there are $n$, suffice.

## K   API USAGE FOR MODEL QUERIES

We report a few important notes on the API calls made to OpenAI models for empirical evaluation of our algorithm:

1. A temperature of 1.0 was used to generate alternate responses for frequency scoring; a temperature of 0.0 was used for all other API calls.
2. GPT-4 was used for the generation of outputs for the MATH questions.
3. GPT-4 was used for self-consistency scoring, described in Section 4.
4. GPT-4o was used for graph generation.

### K.1   DEPENDENCY GRAPH GENERATION PROMPT (MATH/FELM)

**GPT-4o**   Our prompt for graph generation includes in-context exemplars annotated with rationales ("commentary") for guided decomposition of the model-generated output into claims and their relation to one another.

---

I'm going to give you a question and a series of claims in response to the question. I want you to create a dependency graph to represent the relationships between claims. The set of vertices should be the set of claims. Then, if a claim "a" relies on another claim "b" to be considered true, include edge (b, a) in the graph (so a node's ancestors should contain all of its necessary assumptions). Vertices that are "a priori" (e.g., assumptions given in the question, definitions, etc.), should not have ancestors. Your final output will be an adjacency list.

Next, I'll give you some examples to make this clear.

Question: How many vertical asymptotes does the graph of $y = \frac{x}{x^2+1}$ have?

claim 1: A function has vertical asymptotes exactly where its denominator equals zero. claim 2: To solve for the vertical asymptotes of the function $y = \frac{x}{x^2+1}$, we therefore must solve $x^2 + 1 = 0$. claim 3: For all real values of $x$, $x^2 + 1 > 0$ claim 4: Thus, we conclude that the function $y = \frac{x}{x^2+1}$ has no vertical asymptotes.

Desired Output: $[[0, 0, 0, 0], [1, 0, 0, 0], [0, 1, 0, 0], [0, 1, 1, 0]]$

Commentary:

You should output an object like the one above without any other reasoning or formatting. In particular, you should output an array of n arrays, each of length n, where n is the number of claims. If claim j relies on the information from claim i, the jth array should have the ith entry = 1; otherwise this entry should be zero. In this case, note that claim 1 does not have ancestors,

---

because it does not require other steps to be justified (we assume common mathematical theorems, like the presence of vertical asymptotes when the denominator is zero, to be a priori). However, claim 2 relies on the conclusion of claim 1 since it sets the denominator equal to zero. claim 3 implicitly relies on claim 2, since we derive this check from claim 2. Also, the final answer, claim 4, relies on combining information from both claims 2 and 3 (which describe the significance of the equation $x^2 + 1 = 0$ and its answer, respectively). Also note that in generating this graph, we represent implicit relationships between claims: claim 4, for instance, does not cite claims 2 and 3 explicitly, but it certainly relies on their contents. For this reason, we put those edges in its adjacency list. It is very important to represent all relationships in this way. In general, it is unlikely that a claim should be completely "floating" (not relied upon by or reliant upon another claim); in this case, it would not be contributing to the complete output.

By convention, we never include a claim in its own adjacency list (we do not consider a claim to rely on itself).

Here, we're interested in the dependency between claims, not just the correctness. For this reason, it's also important to represent these dependencies even in the case that an answer is wrong.

I'll give you another example below.

Question: Consider the function $y = x^2 + 2x + 15$. What is the sum of the zeroes of this function?

claim 1: The zeroes of a function are the x-values of its x-intercepts. claim 2: To find the zeroes of $y = x^2 + 2x + 15$, we set the right hand side equal to 0, writing $0 = x^2 + 2x + 15$. claim 3: To solve $0 = x^2 + 2x + 15$, we factor it as $0 = (x + 3)(x - 5)$. claim 4: This means that the zeroes of $y = x^2 + 2x + 15$ are $x = -3, 5$. claim 5: We conclude that the sum of the zeroes of this function is $-3 + 5 = 2$.

Desired Output: $[[0, 0, 0, 0, 0], [1, 0, 0, 0, 0], [0, 1, 0, 0, 0], [0, 0, 1, 0, 0], [0, 0, 0, 1, 0]]$

Commentary:

Here, each claim simply relies on the previous claim. Importantly, claim 3 makes an algebraic error, incorrectly factoring as $0 = (x + 3)(x - 5)$ instead of $0 = (x - 3)(x + 5)$, which means the final answer is wrong. Even so, this claim relies on information from claim 2, and claim 4 relies on the conclusion from claim 3, so we represent these edges in our final output. We are agnostic to correctness, and focus solely on the reliance between claims. If claim i makes use of claim j, even incorrectly, claim j should be an ancestor of claim i in our adjacency list.

Now, I'm going to give you another question and list of claims, as before. With all of this explanation in mind, I want you to output an adjacency list with no other reasoning.

**Llama-3.1-70B-Instruct**  Llama had more difficulty with this task, especially replicating the dimensions of the adjacency list, so we reworked the few-shot prompt and gave more explicit instruction. Despite our best efforts, it occasionally output cyclic graphs, in which case we simply considered the trivial "linear" graph ($1 \rightsquigarrow 2 \rightsquigarrow ... \rightsquigarrow n$); our empirical results suggest that, while imperfect, its graphs were still useful.

You are a system designed to create dependency graphs for subclaims in response to a given question. Your output must strictly adhere to the following instructions:

1. Graph Description:
- Represent the dependency relationships between subclaims as a directed graph.
- Each subclaim is a vertex in the graph.
- An edge ($b \rightarrow a$) exists if subclaim "$a$" depends on subclaim "$b$."
- Subclaims that are "a priori" (e.g., assumptions or definitions) should not have any ancestors.

2. Output Format:
- Provide your graph as an adjacency list of size NUM × NUM, where NUM is the number of subclaims (this will be given at the beginning of the prompt).

- Each entry in the adjacency list is a list of n integers:
- A value of 1 at position $i$ in row $j$ indicates that subclaim $j$ depends on subclaim $i$. - A value of 0 indicates no dependency. - Ensure no claim depends on itself (diagonal entries must be 0).

3. Rules: - The adjacency list must be square, with $n$ rows and $n$ columns, where $n$ is the exact number of subclaims provided. - Each row and column must be exactly $n$ integers. Do not include extra rows, columns, or misaligned entries. - The output must consist solely of the adjacency list (e.g., $[[0, 1, 0], [0, 0, 1], [0, 0, 0]]$); do not include explanations, commentary, or any other formatting.

4. Dependencies:
- Consider explicit and implicit dependencies between subclaims. For example, if subclaim $j$ implicitly relies on subclaim $i$ (even if not stated directly), include the edge $(i \rightarrow j)$ in the graph.
- Always represent dependencies, even if the subclaims are incorrect or contain logical errors.

Examples:

- Input:
Question: How many vertical asymptotes does the graph of $y = x/(x^2 + 1)$ have?

NUM = 4 Subclaims:
1. A function has vertical asymptotes exactly where its denominator equals zero.
2. To solve for the vertical asymptotes of the function $y = x/(x^2+1)$, we therefore must solve $x^2 + 1 = 0.3$. For all real values of $x$, $x^2 + 1 > 0$.
4. Thus, we conclude that the function $y = x/(x^2 + 1)$ has no vertical asymptotes.

Desired Output: [[0,0,0,0],[1,0,0,0],[0,1,0,0],[0,1,1,0]]

- Input:
Question: Consider the function $y = x^2 + 2x + 15$. What is the sum of the zeroes of this function?

NUM = 5
Subclaims:
1. The zeroes of a function are the $x$-values of its $x$-intercepts.
2. To find the zeroes of $y = x^2 + 2x + 15$, we set the right-hand side equal to 0, writing $0 = x^2 + 2x + 15$.
3. To solve $0 = x^2 + 2x + 15$, we factor it as $0 = (x + 3)(x - 5)$.
4. This means that the zeroes of $y = x^2 + 2x + 15$ are $x = -3, 5$.
5. We conclude that the sum of the zeroes of this function is $-3 + 5 = 2$.

Desired Output: [[0,0,0,0,0],[1,0,0,0,0],[0,1,0,0,0],[0,0,1,0,0],[0,0,0,1,0]]

Now provide your adjacency list for the following question and subclaims:

## K.2 Self-consistency (frequency) scoring prompt

You will get a list of claims and piece of text. For each claim, score whether the text supports, contradicts, or is unrelated to the claim. Directly return a jsonl, where each line is {"id":[CLAIM_ID], "score":[SCORE]}. Directly return the jsonl with no explanation or other formatting. For the [SCORE], return 1 for supports, $-1$ for contradicts, and 0 for unrelated. The claims are: {CLAIMS}

## K.3 Re-prompting with Filtered Output Prompt

I am going to give you a question some starter work. Please fill in the starter work to provide a complete answer to the question. Question: [QUESTION], Starter Work: [STARTER_WORK]

## L    COMPARING OUTPUTS GENERATED FROM INDEPENDENT FACTUALITY AND COHERENT FACTUALITY

### L.1    MATH DATASET OUTPUTS

---

**Example 1**

**Question:** Solve for $w$ and express as a common fraction: $1\frac{1}{6w} = \frac{42}{3}$

**Independent Factuality:**

1. Multiplying both sides by $w$ gives $1\frac{1}{6} = w \cdot \frac{42}{3}$.

2. Converting $1\frac{1}{6}$ to an improper fraction gives $\frac{7}{6} = 14w$.

**Coherent Factuality:**

1. Multiplying both sides by $w$ gives $1\frac{1}{6} = w \cdot \frac{42}{3}$.

2. Simplifying the right side gives $1\frac{1}{6} = 14w$.

3. Converting $1\frac{1}{6}$ to an improper fraction gives $\frac{7}{6} = 14w$.

---

---

### Example 2

**Question:** Mr. Madoff invests 1000 dollars in a fund that compounds annually at a constant interest rate. After three years, his investment has grown to 1225 dollars. What is the annual interest rate, as a percentage? (Round your answer to the nearest integer.)

**Independent Factuality:**

1. In the formula, $A$ is the amount of money accumulated after n years, including interest.

2. In the formula, $P$ is the principal amount (the initial amount of money).

3. In the formula, $r$ is the annual interest rate (in decimal).

4. In this case, $A = 1225$, $P = 1000$, $n = 1$, and $t = 3$ years.

5. Substituting the given values into the formula, we get: $1225 = 1000(1 + r/1)^{1*3}$.

6. Simplifying the equation, we get: $1225 = 1000(1 + r)^3$

7. Dividing both sides by 1000, we get: $1.225 = (1 + r)^3$

8. Taking the cube root of both sides, we get: $1.07 = 1 + r$.

9. Subtracting 1 from both sides, we get: $r = 0.07$.

10. The annual interest rate is 0.07, or 7% when expressed as a percentage.

11. Rounded to the nearest integer, the annual interest rate is 7%.

**Coherent Factuality:**

1. The formula for compound interest is $A = P(1 + r/n)^{nt}$.

2. In the formula, $A$ is the amount of money accumulated after n years, including interest.

3. In the formula, $P$ is the principal amount (the initial amount of money).

4. In the formula, $r$ is the annual interest rate (in decimal).

5. In the formula, $n$ is the number of times that interest is compounded per year.

6. In this case, $A = 1225$, $P = 1000$, $n = 1$, and $t = 3$ years.

7. Substituting the given values into the formula, we get: $1225 = 1000(1 + r/1)^{1*3}$.

8. Simplifying the equation, we get: $1225 = 1000(1 + r)^3$.

9. Dividing both sides by 1000, we get: $1.225 = (1 + r)^3$.

10. Taking the cube root of both sides, we get: $1.07 = 1 + r$.

11. Subtracting 1 from both sides, we get: $r = 0.07$.

12. The annual interest rate is 0.07, or 7% when expressed as a percentage.

13. Rounded to the nearest integer, the annual interest rate is 7%.

---

### Example 3

**Question:** What is the smallest real number $x$ in the domain of the function

$$g(x) = \sqrt{(x - 3)^2 - (x - 8)^2} \ ?$$

**Independent Factuality:**

1. The expression under the square root must be nonnegative.

2. We must have $(x - 3)^2 - (x - 8)^2 \geq 0$.

3. This simplifies to $(x - 3 + x - 8)(x - 3 - (x - 8)) \geq 0$.

4. The smallest such $x$ is $\boxed{\dfrac{11}{2}}$.

**Coherent Factuality:**

1. The expression under the square root must be nonnegative.

2. We must have $(x - 3)^2 - (x - 8)^2 \geq 0$.

3. This simplifies to $(x - 3 + x - 8)(x - 3 - (x - 8)) \geq 0$.

4. This further simplifies to $(2x - 11)(5) \geq 0$.

**Example 4**

**Question:** If $(2x + 5)(x - 3) = 14$, find the sum of the possible values of $x$.

**Independent Factuality:**

1. The expanded equation simplifies to $2x^2 - x - 29 = 0$

2. Vieta's formulas are used to find the sum of the roots of the equation

3. The sum of the roots of the equation is $\frac{1}{2}$

**Coherent Factuality:**

1. The left side of the equation is expanded to $2x^2 - 6x + 5x - 15 = 14$

2. The expanded equation simplifies to $2x^2 - x - 29 = 0$

3. Vieta's formulas are used to find the sum of the roots of the equation

4. The sum of the roots of the equation is $\frac{1}{2}$

**Example 5**

**Question:** The points $(x, y)$ represented in this table lie on a straight line. The point $(13, q)$ lies on the same line. What is the value of $p + q$? Express your answer as a decimal to the nearest tenth.

| $x$ | $y$ |
|---|---|
| 2 | $-5$ |
| $p$ | $-14$ |
| $p + 2$ | $-17$ |

.

**Independent Factuality:**

1. The slope of the line is calculated as $\frac{-17 - (-14)}{p+2-p} = \frac{-3}{2}$.

2. The slope of the line is also calculated as $\frac{q - (-5)}{13 - 2} = \frac{q+5}{11}$.

3. The slope of the line is also calculated as $\frac{-14 - (-5)}{p - 2} = \frac{-9}{p-2}$.

4. Setting this slope equation equal to $\frac{-3}{2}$ gives $\frac{-3}{2} = \frac{q+5}{11}$.

5. Setting these two slope equations equal gives $\frac{-3}{2} = \frac{-9}{p-2}$.

**Coherent Factuality:**

1. The slope of the line is calculated as $\frac{-17 - (-14)}{p+2-p} = \frac{-3}{2}$.

2. Setting these two slope equations equal gives $\frac{-3}{2} = \frac{-9}{p-2}$.

3. The slope of the line is also calculated as $\frac{q - (-5)}{13 - 2} = \frac{q+5}{11}$.

4. Setting this slope equation equal to $\frac{-3}{2}$ gives $\frac{-3}{2} = \frac{q+5}{11}$.

## L.2 FELM DATASET OUTPUTS

**Example 1**

**Question:** Jessica makes $2,000.00 a month. She sets 25% of her paycheck aside to put towards fancy shoes. Each pair of shoes she buys costs $1,000.00. How many shoes can she buy in a year?

**Independent Factuality:**

1. Jessica sets aside 25% of her paycheck, which is: $2,000.00 x 0.25 = $500.00

2. So Jessica can buy 6 pairs of shoes in a year with the money she sets aside from her paycheck.

**Coherent Factuality:**

1. Jessica sets aside 25% of her paycheck, which is: $2,000.00 x 0.25 = $500.00

2. To figure out how many pairs of shoes she can buy in a year, we need to multiply the number of pairs she can buy in a month by 12 (the number of months in a year):$500.00 x 12 = $6,000.00.

---

### Example 2

**Question:** The ratio of men to women on a bus is 5:9. If the total number of passengers on the bus is 84, and 20 women alight from the bus at the next stop, how many women have remained on the bus?

**Independent Factuality:**

1. If the total number of passengers on the bus is 84, then each ratio unit represents $\frac{84}{14} = 6$ passengers.

2. The number of women on the bus is 9 units $\times$ 6 passengers/unit = 54 passengers.

3. If 20 women alight from the bus, then the number of women remaining on the bus is $54 - 20 = 34$ women. Answer: $\boxed{34}$.

**Coherent Factuality:**

1. The total ratio of men and women on the bus is $5 + 9 = 14$.

2. If the total number of passengers on the bus is 84, then each ratio unit represents $\frac{84}{14} = 6$ passengers.

3. The number of women on the bus is 9 units $\times$ 6 passengers/unit = 54 passengers.

4. If 20 women alight from the bus, then the number of women remaining on the bus is $54 - 20 = 34$ women. Answer: $\boxed{34}$.

---

### Example 3

**Question:** John injured his back and needs to go to physical therapy. He went to physical therapy for 6 weeks. Each week he went twice for 2 hours at a time. If the sessions cost $125 per hour how much did the physical therapy cost?

**Independent Factuality:**

1. He went to therapy $6 \times 2 = \ll 6 \times 2 = 12 \gg 12$ times.

2. So he paid $250 \times 12 = \$ \ll 250 \times 12 = 3000 \gg 3000$.

3. Answer: $\boxed{3000}$.

**Coherent Factuality:**

1. He went to therapy $6 \times 2 = \ll 6 \times 2 = 12 \gg 12$ times.

2. Each session cost $2 \times 125 = \$ \ll 2 \times 125 = 250 \gg 250$.

3. So he paid $250 \times 12 = \$ \ll 250 \times 12 = 3000 \gg 3000$.

4. Answer: $\boxed{3000}$.

## M  INDEPENDENT FACTUALITY BASELINES

We also report the baseline results of independent factuality as in (Mohri & Hashimoto, 2024) for the problems we analyze; these plots are analogous to those we report in Section 6.

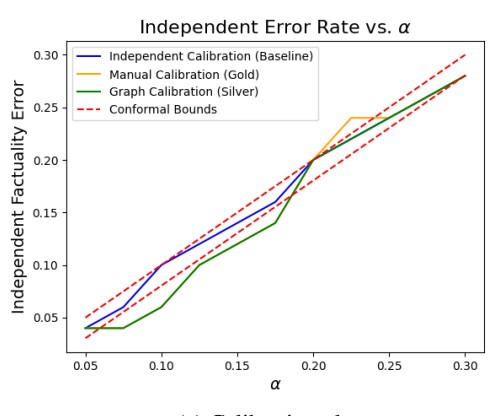

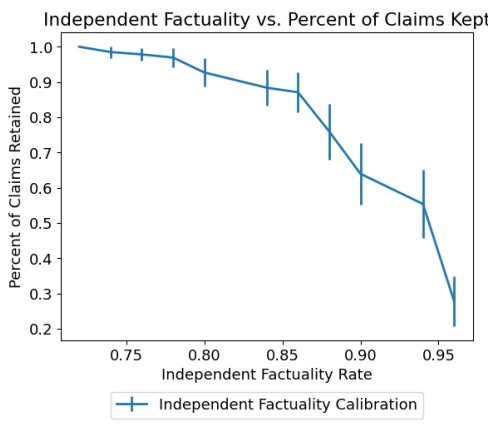

(a) Calibration plot

(b) Percent of claims kept vs. independent factuality

Figure 8: These figures depict the performance of independent factuality calibration validated against independent factuality. We can see that the calibration guarantees still hold and useful proportions of claim retention, however, as claims may still be retained despite claims that preceding it being deemed as incorrect, this does not reflect our coherent definition of factuality.

## N  LEGIBILITY RESULTS

To measure legibility, we asked GPT-4o and Llama-3.1-70B-Instruct to grade outputs as erroneous or factual.

All queries were at temperature = 0. We considered all outputs across $\alpha = 0.1, 0.15, 0.2$ for which (1) our method and the baseline produced different, non-empty outputs and (2) both outputs had the same independent factuality (both contained a hallucination or both didn't). The task was error detection, so "false positive" means GPT graded an output as containing an error when it didn't.

1. **GPT-4 outputs, GPT-4o as judge**

   Table 2: Results of legibility experiment with LLM-as-a-judge with our method.

   | Outcome | Proportion |
   | --- | --- |
   | True Positive | 0.22 |
   | True Negative | 0.59 |
   | False Positive | 0.17 |
   | False Negative | 0.02 |

   Table 3: Results of legibility experiment with LLM-as-a-judge with the baseline, independent factuality method.

   | Outcome | Proportion |
   | --- | --- |
   | True Positive | 0.17 |
   | True Negative | 0.46 |
   | False Positive | 0.32 |
   | False Negative | 0.05 |

2. **GPT-4 outputs, Llama-3.1-70B-Instruct as judge**

Table 4: Results of legibility experiment with LLM-as-a-judge with our method.

| Outcome | Proportion |
| --- | --- |
| True Positive | 0.15 |
| True Negative | 0.61 |
| False Positive | 0.15 |
| False Negative | 0.10 |

Table 5: Results of legibility experiment with LLM-as-a-judge with the baseline, independent factuality method.

| Outcome | Proportion |
| --- | --- |
| True Positive | 0.10 |
| True Negative | 0.54 |
| False Positive | 0.24 |
| False Negative | 0.12 |

3. **Llama-3.1-70B-Instruct outputs, GPT-4o as judge**

Table 6: Results of legibility experiment with LLM-as-a-judge with our method.

| Outcome | Proportion |
| --- | --- |
| True Positive | 0.08 |
| True Negative | 0.64 |
| False Positive | 0.26 |
| False Negative | 0.03 |

Table 7: Results of legibility experiment with LLM-as-a-judge with the baseline, independent factuality method.

| Outcome | Proportion |
| --- | --- |
| True Positive | 0.06 |
| True Negative | 0.53 |
| False Positive | 0.36 |
| False Negative | 0.05 |

4. **Llama-3.1-70B-Instruct outputs, Llama-3.1-70B-Instruct as judge**

Table 8: Results of legibility experiment with LLM-as-a-judge with our method.

| Outcome | Proportion |
| --- | --- |
| True Positive | 0.03 |
| True Negative | 0.83 |
| False Positive | 0.08 |
| False Negative | 0.08 |

Table 9: Results of legibility experiment with LLM-as-a-judge with the baseline, independent factuality method.

| Outcome | Proportion |
|---|---|
| True Positive | 0.03 |
| True Negative | 0.78 |
| False Positive | 0.11 |
| False Negative | 0.09 |

Grading Prompt:

Given the prompt: [prompt]

Do the following subclaims contain any errors?

[Subclaims separated by newlines]

Respond with exactly one character: either 'Y' for 'there's an error' or 'N' for 'no error.'

