# OpenReview forum: "Conformal Language Model Reasoning with Coherent Factuality"
_ICLR.cc/2025/Conference — ICLR 2025 Poster_

### Official Review · Reviewer_D79W · 2024-10-30

**Soundness:** 2
**Presentation:** 2
**Contribution:** 2
**Rating:** 6
**Confidence:** 4

**Summary:**

This work presents a conformal prediction framework for LLMs on reasoning tasks. The key difference between this work and previous work is the consideration of dependencies between claims. Unlike previous framework that scores and removes each claim independently, the proposed framework generates graph with each node representing a claim in the response, and then score and remove the claims while considering the graph structure. On MATH and FELM, the proposed method shows better calibration and stronger guarantee compared to previous methods and a few simpler baselines.

**Strengths:**

1. Extending the conformal prediction framework to reasoning problems is an important direction. The idea of considering the dependency structure among the claims is straightforward and effective.
2. The proposed framework is simple to implement and shows stronger performance than baselines in the experiments.

**Weaknesses:**

1. This writing of this paper can be substantially improved and in general should be more rigorous. There are a number of writing issue in this paper making this paper a bit hard to understand. To list a few:
    * One key property of the ideal graph is it uses the "minimal set of the claims". However, this is only mentioned in the appendix.
	* What is the "graph G" at LINE 350? Is it the corresponding subgraph to each node?
	* LINE 402 points to appendix F, but appendix F does not contain the prompts.
	* Figure 4(b) and 4(c) use the same title. This is confusing and seems to be typos.
	* What is "Descendants weight boosting" in LINE 501?
	* What is "independently filtered outputs" in LINE 509?
	* What is "self-consistency scoring" in LINE 970?
	* I don't understand the sentence from LINE 316-318.
	* The graph generation step is a critical part of the proposed method, but all the details are in the appendix. I can understand the specific prompt to be in the appendix, but there need to be some high-level descriptions in the main paper.
2. Reasoning problems are different from general fact-related questions as they often-times require a single correct conclusion. While it's still useful to provide partially correct responses, it is important in this case to also report (e.g., through human studies) how many of the responses actually contain the correct answer, or how useful these responses are after filtering.
3. The proposed method is only tested on one model (which I believe is GPT-4o, but the paper does not explicitly mention where the model responses on MATH come from). It would be great to test at least one more model to see how generalizable the proposed framework is. If the authors can use open-source models, it will also greatly improve the reproducibility of this paper.

**Questions:**

1. Why do you use "median" in LINE 361? And how do you select the hyper-parameter $\beta$?
2. Do you make any special design to deal with the cases in LINE 153 (i.e., "it is not reasonable to do so in a proof of that fact")?

---

> ### Author Response · Authors · 2024-11-22
> **Response to Reviewer D79W (Part 1/2)**
>
> We thank Reviewer D79W for taking the time to review our paper, and for their helpful suggestions. We address the points raised in the review below:
>
> **Weakness 1**
>
> Thank you for the suggestions on writing-related revisions! We have incorporated this feedback into our revised version, which we hope clarifies some terminology that was unclear before and addresses the concerns raised.
>
> > One key property of the ideal graph is it uses the “minimal set of the claims”. However, this is only mentioned in the appendix.
>
> We have updated the main text to include a note on the minimal set of claims.
>
> > What is the “graph G” at LINE 350? Is it the corresponding subgraph to each node?
>
> Graph G refers to the approximate deducibility graph as defined in Definition 4, whose subgraphs we examine.
>
> >LINE 402 points to Appendix F, but appendix F does not contain the prompts.
>
> Thank you for pointing this out -- this has been corrected to point to Appendix K, which does contain the prompts.
>
> > Figure 4(b) and 4(c) use the same title. This is confusing and seems to be typos.
>
> Thank you for bringing this to our attention -- we’ve corrected this in the revised version.
>
> > What is “Descendants weight boosting” in LINE 501?
>
> “Descendants weight boosting” refers to the “Descendant Weighting” scoring function introduced in Section 4. We have reworded this accordingly in line 503 of our revised paper.
>
> > What is “independently filtered outputs” in LINE 509?
>
> “Independently filtered outputs” refers to the application of Mohri and Hashimoto, 2024’s method of treating claims as independent of one another, which we termed “independent factuality”.
>
> > What is “self-consistency scoring” in LINE 970?
>
> Self-consistency scoring is a frequency score measure introduced in Mohri and Hashimoto, 2024, wherein several (e.g. 5) additional responses are sampled from the model for a given prompt. For each generation, the model determines whether the claim supports, contradicts, or is independent of the target claim, assigning +1, -1, or 0 to that output, and yielding a score for the target claim in [-5, 5]. We include a similar  description of this approach in Section 4, in the discussion about the scoring functions used in our algorithm.
>
> > I don't understand the sentence from LINE 316-318.
>
> This sentence refers to the notion that GPT-4o-generated approximate deducibility graphs may not be minimal, and since we simply require them to have sufficient substantiation sets, they are constructed in such a manner that they may contain more edges than would be needed in the ideal deducibility graph.
>
> **Weakness 2**
>
> > While it's still useful to provide partially correct responses, it is important in this case to also report (e.g., through human studies) how many of the responses actually contain the correct answer, or how useful these responses are after filtering.
>
> We examine the utility of coherently factual outputs by re-prompting the model conditional on the outputs of our algorithm. This involves generating a coherently factual output through our subgraph filtering protocol (which constitutes a partial reasoning chain) and re-prompting the model to complete the reasoning chain (i.e. filling in any missing steps). These results are in Table 1 of our work, which shows that these outputs can be made iteratively more useful by reducing the factuality error. That us, this bootstrapping method of filtering with our protocol and re-prompting is more effective for coherent factuality than it is for independently factual outputs; for example, with $\\alpha = 0.05$, the factuality error was reduced to 0.10 with bootstrapping on coherently factual responses, as opposed to 0.26 when bootstrapping independently factual responses.

---

> ### Author Response · Authors · 2024-11-22
> **Response to Reviewer D79W (Part 2/2)**
>
> **Weakness 3**
>
> The responses in our experiments are generated from GPT-4, with the proxy graphs being generated from GPT-4o. We have now made explicit mention of this in Section 5, and have produced results with Llama-3.1-70B-Instruct, given the reviewer’s helpful feedback on reproducibility with open-source models. These results are included in Appendix E of our revision.
>
> Our silver-calibrated, validated results were similar to those for GPT. Both conformal bounds were satisfied in the calibration experiment (see Figure 6a). We retain a competitive percentage of claims relative to independent factuality (see below), despite a stricter definition, while attaining an empirical factuality coverage close to the target rate.
>
> | Alpha (Target Factuality | Coherent Factuality (Ours) | Coherent Factuality (Baseline) | Claims Retained (Ours) | Claims Retained (Baseline) |
> | ------------------------ | -------------------------- | ------------------------------ | ---------------------- | -------------------------- |
> | 0.979 | 0.98                 | 0.68                           | 0.38                   | 0.48                       |
> | 0.958                    | 0.96                       | 0.66                           | 0.54                   | 0.49                       |
> | 0.9375                   | 0.94                       | 0.68                           | 0.56                   | 0.61                       |
> | 0.917                    | 0.92                       | 0.66                           | 0.70                   | 0.67                       |
> | 0.896                    | 0.9                        | 0.64                           | 0.73                   | 0.79                       |
> | 0.875                    | 0.88                       | 0.66                           | 0.77                   | 0.82                       |
> | 0.854                    | 0.86                       | 0.64                           | 0.82                   | 0.83                       |
> | 0.833                    | 0.84                       | 0.62                           | 0.86                   | 0.86                       |
> | 0.8125                   | 0.82                       | 0.66                           | 0.87                   | 0.90                       |
> | 0.792                    | 0.8                        | 0.7                            | 0.88                   | 0.93                       |
> | 0.771                    | 0.78                       | 0.68                           | 0.91                   | 0.96                       |
> | 0.750                    | 0.76                       | 0.7                            | 0.95                   | 0.98                       |
>
> We are currently working on validating these results with respect to gold-annotations, which requires more annotations. We will add these to the camera-ready version.
>
> To further aid with the reproducibility of our approach, we have also included the following cost estimates for generating proxy graphs and producing responses, and included them in Appendix J of our revised paper. For each example in the calibration and test set, the algorithm requires 8 queries comprising at most 16k tokens; for our calibration set of 50 examples, this cost in total less than $\\$5.00$ using GPT and less than $\\$0.70$ using Llama; the same queries are made for the test set, so each test example cost less than $\\$0.10$ for GPT and $\\$0.01$ for Llama. These estimates are conservative, assuming full utilization of 2000-token total context and output to accommodate longer response lengths (although our responses were much shorter).
>
> Question 1:
> > Why do you use “median” in LINE 361? And how do you select the hyper-parameter?
>
> We explored several similar graph-sensitive scoring mechanisms, each motivated by weighting the risk score of a node according to the risk scores of its ancestors and/or descendants. This median version seemed most robust in performance to small changes in beta (we speculate this is because the median is not sensitive to outlier scores). We swept beta values in [0, 1] and chose 0.5 for its good performance. This information is now included in the scoring section as a footnote.
>
> Question 2:
>
> > Do you make any special design to deal with the cases in LINE 153 (i.e., “it is not reasonable to do so in a proof of that fact”)?
>
> As our algorithm does not depend on a complete practical instantiation of the ground truth $C_{true}$, this is addressed during annotation on the basis of human understanding of the annotator’s understanding of the priors necessary to solve a given problem (e.g. the math axioms) and interpretation of the given context. This serves as a reasonable proxy, following from prior works such as Mohri and Hashimoto, 2024.
>
> ---
> We hope that our response and revised paper addresses your concerns and questions. Please let us know if you have any further questions, and we would be happy to answer them!

---

> > ### Comment · Reviewer_D79W · 2024-11-24
> >
> > Thank the authors for the detailed response and all the updates in the paper! I have increased my score from 3 to 5 as most of the writing unclarities have been addressed. I still have concerns over weakness 2. Yes, I agree that the responses produced by the proposed method have real utility as reprompting them leads to better responses. However, if you do re-prompting, you lose all the formal guarantees, which I feel is the core point of this paper. So at the end of the day, I still believe a direct evaluation without re-prompting is valuable. Otherwise, I'm curious if combined with re-prompting, what is the advantage of this method compared to all the other prompting methods out there.

---

> > > ### Author Response · Authors · 2024-11-27
> > > **Response to Reviewer D79W (Direct Evaluation: Legibility)**
> > >
> > > Thank you for your detailed feedback!
> > >
> > > > "However, if you do re-prompting, you lose all the formal guarantees, which I feel is the core point of this paper."
> > >
> > > It’s a good point that re-prompting does not retain guarantees, which is one of the primary strengths of our work. We note that Mohri and Hashimoto, 2024 [1] similarly loses guarantees by reprompting to merge filtered outputs (we did not feel a merge step was necessary because a step-wise presentation of math outputs is typical). However, we understand the importance of showing the utility of coherent factuality directly: we find that our method has improved "legibility" (in line with Kirchner et al. 2024 [2]) over the baseline. **Legibility** is the ability of observers to understand and spot errors in an output, so legible outputs are either plainly correct or plainly incorrect, and downstream users can confidently decide when to use them.
> > >
> > > We defer human legibility studies to future work, but as a proxy, we asked GPT-4o and Llama-3.1-70B-Instruct to “grade” filtered outputs (original, not reprompted) as either correct or erroneous (more details in footnote). For each combination of output generation method (GPT, Llama) and output grading method (GPT, Llama), our method was **more legible than the baseline** (lower false positive and false negative rates for fixed levels of factuality). The task was error detection, so, e.g., “false positive” means GPT graded an output as containing an error when it didn’t.
> > >
> > > **1) GPT-4 outputs, GPT-4o as judge:**
> > >
> > > a) *Ours*
> > > | Outcome | Proportion  |
> > > |------|----|
> > > | True Positive | 0.22 |
> > > | True Negative | 0.59  |
> > > | False Positive | 0.17 |
> > > | False Negative | 0.02 |
> > >
> > > b) *Baseline*
> > > | Outcome | Proportion  |
> > > |------|----|
> > > | True Positive | 0.17  |
> > > | True Negative | 0.46  |
> > > | False Positive| 0.32 |
> > > | False Negative| 0.05 |
> > >
> > > **2) GPT-4 outputs, Llama-3.1-70B-Instruct as judge:**
> > >
> > > a) *Ours*
> > > | Outcome | Proportion  |
> > > |------|------|
> > > | True Positive| 0.15  |
> > > | True Negative| 0.61  |
> > > | False Positive |0.15 |
> > > | False Negative| 0.10 |
> > >
> > > b) *Baseline*
> > > | Outcome | Proportion |
> > > |------|-----|
> > > | True Positive | 0.10  |
> > > | True Negative | 0.54  |
> > > | False Positive| 0.24 |
> > > | False Negative| 0.12 |
> > >
> > > **3) Llama-3.1-70B-Instruct outputs, GPT-4o as judge:**
> > >
> > > a) *Ours*
> > > | Outcome | Proportion |
> > > |--------|-----|
> > > | True Positive| 0.08  |
> > > | True Negative | 0.64  |
> > > | False Positive | 0.26 |
> > > | False Negative| 0.03 |
> > >
> > > b) *Baseline*
> > > | Outcome  | Proportion |
> > > |---------|--------|
> > > | True Positive | 0.06 |
> > > | True Negative | 0.53 |
> > > | False Positive | 0.36 |
> > > | False Negative | 0.05 |
> > >
> > > **4) Llama-3.1-70B-Instruct outputs, Llama-3.1-70B-Instruct as judge:**
> > >
> > > a) *Ours*
> > > | Outcome  | Proportion |
> > > |-------|------|
> > > | True Positive| 0.03 |
> > > | True Negative | 0.83 |
> > > | False Positive | 0.08 |
> > > | False Negative| 0.08 |
> > >
> > > b) *Baseline*
> > > | Outcome  | Proportion |
> > > |--------|-----|
> > > | True Positive | 0.03 |
> > > | True Negative | 0.78 |
> > > | False Positive | 0.11 |
> > > | False Negative | 0.09 |
> > >
> > > Of course, it is still important that filtered outputs contain the correct answer. We manually checked all GPT outputs filtered by our method and the baseline at $\\alpha = 0.1.$ Of hallucination-free outputs, **64%** contained the correct final answer, the **same rate as the** (coherence-free, less legible) **baseline**. Note that some were examples on which GPT’s hallucinated solution had been filtered out, and so filtering had no chance of outputting a correct solution. For several outputs which technically did not contain the final answer, computing the answer based on the output’s last step was trivial (e.g., completing a sum or choosing between a positive and negative solution). Our filtered outputs preserve coherence to scaffold users toward correct conclusions while controlling for hallucinations.
> > >
> > > > what is the advantage of this method compared to all the other prompting methods out there
> > >
> > > Relative to other prompting strategies, our method confers the unique benefit of post-hoc calibrated factuality with theoretical guarantees. Since this method expects an arbitrary model, it can be appended to any existing reprompting strategy in order to guarantee factual and coherent outputs.
> > >
> > > **Conclusion**
> > >
> > > Even without re-prompting, our approach improves upon the coherence and legibility of the baseline while retaining its output completeness as measured by percent of claims retained and proportion of outputs with correct final answers.
> > >
> > > **References**
> > >
> > > [1] Christopher Mohri and Tatsunori Hashimoto, “Language Models with Conformal Factuality Guarantees.” arXiv preprint, arXiv:2402.10978 (2024).
> > >
> > > [2] Jan Kirchner et al. “Prover-Verifier Games improve legibility of LLM outputs”. arXiv preprint, arXiv: 2407:13692 (2024).
> > >
> > > ---
> > >
> > > Please let us know if you have any further questions/concerns!

---

> > > > ### Author Response · Authors · 2024-11-27
> > > > **Footnote on Direct Evaluation Details**
> > > >
> > > > We provide some details on the experimental details for direct evaluation on legibility, as introduced above:
> > > > * All responses were sampled at temperature = 0.
> > > > * We considered all outputs across $\\alpha = 0.1,0.15,0.2$ for which (1) our method and the baseline produced different, non-empty outputs and (2) both outputs had the same independent factuality (both contained a hallucination or both didn't).
> > > >
> > > > Please note that we will add these results to the Appendix of our revised paper shortly.

---

> > > > > ### Comment · Reviewer_D79W · 2024-12-02
> > > > >
> > > > > Thank the authors for the further response and the additional results.  These direct evaluations on the utility of the coherent factuality are very important for this paper. Hence, I have increased my score to 6, and I would strongly encourage the authors to provide more details for this analysis in the revised paper.

---

### Official Review · Reviewer_DfTB · 2024-11-01

**Soundness:** 2
**Presentation:** 3
**Contribution:** 3
**Rating:** 6
**Confidence:** 2

**Summary:**

The paper defines “coherent factuality” and develops a conformal-predictionbased method to guarantee coherent factuality of language model outputs for reasoning tasks, where claims need to be substantiated and outputted in a comprehensible order to ensure correctness as well as coherence. In addition, they evaluate the method on MATH and FELM datasets, and verify the effectiveness of the method.

**Strengths:**

Several recent works used conformal prediction to verify the correctness of the generation of LLMs with a strong assumption that the factuality of a claim can be independently evaluated. In order to generalize the method to reasoning domains, where claims need to be substantiated and outputted in a comprehensible order, the paper defines a new notion of factuality ”coherent factuality” and develops a conformal-predictionbased method to guarantee coherent factuality of language model outputs. The paper verified the proposed method on MATH and FELM datasets by comparing the results of the baseline proposed in (Mohri & Hashimoto, 2024).

**Weaknesses:**

It is not clear how to create and use $C_{true}$ in the experiments on MATH and FELM datasets. The paper said in Line 150 “In practice, we might choose some reference like Wikipedia or a math textbook as our ground truth”, however, there is no statements about $C_{true}$ in the experiments.

The paper uses GPT4o to generate the graphs, but the quality of the graphs is unknown.

In addition, the proposed method can obtain both coherent factuality and independent factuality of the LLM output, however, there is no experiment to demonstrate whether there is an impact on the performance of downstream tasks. Or can the proposed method improve the performance of the downstream tasks?

**Questions:**

see the weaknesses.

---

> ### Author Response · Authors · 2024-11-22
> **Response to Reviewer DfTB**
>
> We thank Reviewer DfTB for taking the time to review our paper, and for their positive review. Please see below for our responses to the reviewer’s feedback:
>
> > ​​It is not clear how to create and use $C_{true}$ in the experiments on MATH and FELM datasets.
>
> Notably, our algorithm does not directly require knowledge of $C_{true}$ or a complete fixed instantiation of the ground truth in practice, although our guarantees depend on the quality of annotations in the calibration set, which are with respect to $C_{true}$. In the annotation phase, we relied on the annotators’ understanding of the required prior knowledge and the given context of the problem to serve as a reasonable proxy, as in Mohri and Hashimoto, 2024 [1]; for math problems, however, we can reasonably believe that the annotators’ conceptions will be uniform, with similar levels of mathematical knowledge and depth.
>
> > The paper uses GPT4o to generate the graphs, but the quality of the graphs is unknown.
>
> We examined the quality of these graphs against ideal human-annotated graphs; the following is repeated from the general response: We compare human-annotated ideal graphs for the first ten examples against the GPT-4o-generated graphs for the same samples. The edit distance to the ideal deducibility graph was on average 1.8; the edit distance to an approximate deducibility graph was 0 (meaning each graph considered satisfied Definition 4, which is all that is necessary for both bounds to hold). Both methods were calibrated, so coherent factuality was approximately the target $1 -\\alpha$ in either case. Thus, we only include retention results in this table. Note that 1.0 because the baseline accuracy here was 70% (no filtering needed at $\\alpha = 0.3$)
>
> | $\\alpha$ | Claim Retention (Human-Generated Graphs) | Claim Retention (GPT-Generated Graphs) |
> | --------- | ---------------------------------------- | -------------------------------------- |
> | 0.1       | 0.33                                     | 0.33                                   |
> | 0.2       | 0.74                                     | 0.86                                   |
> | 0.3       | 1.0                                      | 1.0                                    |
>
> The plots of the results which include the realized coherent factuality for each of these settings are in Appendix F.
>
> > In addition, the proposed method can obtain both coherent factuality and independent factuality of the LLM output, however, there is no experiment to demonstrate whether there is an impact on the performance of downstream tasks. Or can the proposed method improve the performance of the downstream tasks?
>
> Our results in Table 1 by re-prompting to condition on the previous coherently factual output and fill in the remainder of the reasoning chain addresses the practical viability of our method in yielding useful responses. The motivation for this is that a coherent response is easier to complete than a non-coherent response. Notably, this approach results in decreased factuality error, showing that our empirically achieved factuality does improve via re-prompting and reinforcing our hypothesis; for $\\alpha = 0.05$, the factuality error was reduced to 0.10 with bootstrapping on coherently factual responses, as opposed to 0.26 when bootstrapping independently factual responses.
>
> **References**
>
> [1] Christopher Mohri and Tatsunori Hashimoto, “Language Models with Conformal Factuality Guarantees.” arXiv preprint, arXiv:2402.10978 (2024).
>
> ---
>
> We hope that these responses address your concerns. Please let us know if you have any further questions!

---

> > ### Comment · Reviewer_DfTB · 2024-11-25
> >
> > Thanks for your detailed responses, but I still have a concern about the the impact on the performance of downstream tasks. For example, does the method can improve the final reasoning performance on the MATH and FELM, not just the change in factuality on questions?

---

> ### Author Response · Authors · 2024-11-27
> **Response to Reviewer DfTB (Downstream Utility)**
>
> Thank you for your feedback! You raise a good point: in addition to the factuality of outputs, their downstream utility is important.
> It is first important to note that the primary aim of this work is to detect and filter hallucinations at a calibrated rate while preserving reasoning integrity. We do not consider augmented prompting of the original output (in line with Mohri and Hashimoto, 2024 [1]), so this method does not allow a model to solve a problem it previously couldn’t solve. However, this framework considers outputs from an arbitrary model, so it can be appended to any existing augmented prompting strategy in order to guarantee factual and coherent outputs.
>
> It is still important that we compete with the utility of filtering baselines. We compare the downstream utility of our method with the baseline according to two metrics: legibility of outputs (how transparently true/false they appear) and the rate at which outputs contain a correct answer.
>
> For fixed levels of factuality, our method’s outputs are more legible than baseline filtered outputs. Legibility (Kirchner et al. 2024 [2]) is the ability of observers to understand and spot errors in an output, so legible outputs are either plainly correct or plainly incorrect, and downstream users can confidently decide when to use them.
>
> We defer human legibility studies to future work, but as a proxy, we asked GPT-4o and Llama-3.1-70B-Instruct to “grade” filtered outputs (original, not re-prompted) as either correct or erroneous (more details in footnote). For each combination of output generation method (GPT, Llama) and output grading method (GPT, Llama), our method was more legible than the baseline (lower false positive and false negative rates for fixed levels of factuality). The task was error detection, so, e.g., “false positive” means GPT graded an output as containing an error when it didn’t.
>
> **1) GPT-4 outputs, GPT-4o as judge:**
>
> a) *Ours*
> | Outcome  | Proportion  |
> |------|--------|
> | True Positive | 0.22 |
> | True Negative | 0.59  |
> | False Positive | 0.17 |
> | False Negative | 0.02 |
>
> b) *Baseline*
> | Outcome | Proportion  |
> |------|--------|
> | True Positive | 0.17  |
> | True Negative | 0.46  |
> | False Positive| 0.32 |
> | False Negative| 0.05 |
>
> **2) GPT-4 outputs, Llama-3.1-70B-Instruct as judge:**
>
> a) *Ours*
> | Outcome | Proportion  |
> |------|--------|
> | True Positive| 0.15  |
> | True Negative| 0.61  |
> | False Positive |0.15 |
> | False Negative| 0.10 |
>
> b) *Baseline*
> | Outcome | Proportion  |
> |------|-----|
> | True Positive | 0.10  |
> | True Negative | 0.54  |
> | False Positive| 0.24 |
> | False Negative| 0.12 |
>
> **3) Llama-3.1-70B-Instruct outputs, GPT-4o as judge:**
>
> a) *Ours*
> | Outcome  | Proportion  |
> |------|------|
> | True Positive| 0.08  |
> | True Negative | 0.64  |
> | False Positive | 0.26 |
> | False Negative| 0.03 |
>
> b) *Baseline*
> | Outcome   | Proportion  |
> |------|------|
> | True Positive | 0.06  |
> | True Negative | 0.53  |
> | False Positive | 0.36 |
> | False Negative | 0.05 |
>
> **4) Llama-3.1-70B-Instruct outputs, Llama-3.1-70B-Instruct as judge:**
>
> a) *Ours*
> | Outcome  | Proportion  |
> |------|------|
> | True Positive| 0.03  |
> | True Negative | 0.83  |
> | False Positive | 0.08 |
> | False Negative| 0.08 |
>
> b) *Baseline*
> | Outcome  | Proportion  |
> |-----|-----|
> | True Positive | 0.03  |
> | True Negative | 0.78  |
> | False Positive | 0.11 |
> | False Negative | 0.09 |
>
> Of course, it is still important that filtered outputs contain the correct answer. We manually checked all GPT outputs filtered by our method and the baseline at $\\alpha = 0.1$. Of hallucination-free outputs, **64%** contained the correct final answer, the **same rate as the** (coherence-free, less legible) **baseline**. Note that some were examples on which GPT’s hallucinated solution had been filtered out, and so filtering had no chance of outputting a correct solution.
>
> For several outputs which technically did not contain the final answer, computing the answer based on the output’s last step was trivial (e.g., completing a sum or choosing between a positive and negative solution). Our filtered outputs preserve coherence to scaffold users toward correct conclusions while controlling for hallucinations. Our approach improves upon the coherence and legibility of the baseline while retaining its output completeness as measured by percent of claims retained and proportion of outputs with correct final answers.
>
> *Footnote (Legibility Details)*:
> All queries at temp. = 0. We considered all outputs across $\\alpha = 0.1, 0.15, 0.2$ for which (1) our method and the baseline produced different, non-empty outputs and (2) both outputs had the same independent factuality (both contained a hallucination or both didn't).
>
> [1] Christopher Mohri and Tatsunori Hashimoto, “Language Models with Conformal Factuality Guarantees.” arXiv preprint, arXiv:2402.10978 (2024).
>
> [2] Jan Kirchner et al. “Prover-Verifier Games improve legibility of LLM outputs”. arXiv preprint, arXiv: 2407:13692 (2024).

---

### Official Review · Reviewer_wbML · 2024-11-04

**Soundness:** 3
**Presentation:** 3
**Contribution:** 2
**Rating:** 6
**Confidence:** 3

**Summary:**

The paper defines “coherent factuality” of language model for reasoning tasks and applies a conformal prediction to guarantee coherent factuality. In addition to the split conformal prediction proposed by (Mohri & Hashimoto, 2024), this work proposes a deducibility graph by employing the “deducibility” property instead of “partial entailment” to take in claims by the ground truth. This criticizes that the previous work focuses solely on independent factuality, which makes the strong assumption that sub-claims are independent. Coherence factuality is applied to mathematical reasoning problems such as MATH or FELM, filtering the sub-graphs with the desired coherence factuality.

**Strengths:**

This paper clearly points out that existing conformal factuality is not appropriate for reasoning tasks, and suggests deducibility graph and conformal prediction with coherence factuality. In addition, this experimentally achieved the desired correction and substantiation by applying conformal prediction to the newly defined coherence factuality.
This also proposes a claim-scoring function that considers the graph and reflects the confidence along descendants well.

**Weaknesses:**

It seems to be sufficiently appealed that coherence factuality is more necessary for reasoning tasks than independent factuality. However, if bad claims, as mentioned in the sentence below, are accepted because they are consistent, wouldn't that be of no help in resolving hallucination? I think additional explanations about coherence factuality or deducibility more than as defined in the paper.
"Our definition of deducibility graphs permits the arbitrary treatment of claims that do not follow from the
ground truth".
Similarly, all qualitative results were drawn from the MATH dataset, which has only true claims as far as I know. It appears that additional qualitative results that include bad claims with coherence factuality are needed.

Additionally, an approximate deducibility graph is obtained by creating graph proxies using GPT-4o, but this does not provide theoretical guarantee, which is also mentioned in the paper. This paper said that these graph proxies provide a benefit in imposing the property called dependency, but as mentioned above, it does not come as a big advantage if bad claims are considered, so it appears that the theoretical guarantee of conformal prediction is not fully utilized.

**Questions:**

As mentioned above, are there any results from experiments using a human annotated ideal graph other than GPT-4o?

Although it seems sufficiently argued that coherence factuality is more necessary than independent factuality for reasoning tasks, if bad claims are accepted because they are consistent how would this help in addressing hallucination issues?

Are there extra qualitative results for other datasets (e.g., FELM)?

---

> ### Author Response · Authors · 2024-11-22
> **Response to Reviewer wbML (Part 1/2)**
>
> We thank Reviewer wbML for taking the time to review our paper and for their feedback. We address each of the stated weaknesses below and in our revised draft.
>
> **Handling Bad Claims**
> > However, if bad claims, as mentioned in the sentence below, are accepted because they are consistent, wouldn't that be of no help in resolving hallucination?... Although it seems sufficiently argued that coherence factuality is more necessary than independent factuality for reasoning tasks, if bad claims are accepted because they are consistent how would this help in addressing hallucination issues?
>
> In our calibration algorithm, we consider the lowest score we can filter possible outputs on while still having coherently factual outputs a large proportion of the time. Thus, if bad claims are often consistent, with a high self-consistency score, the calibration algorithm will produce a high threshold with more stringent conditions for including a claim in the final output. Because there is no guarantee on percentage of claims kept, if self-consistency scores have no correlation with the correctness of subclaims, it will remove most claims to retain high correctness.
>
> > Similarly, all qualitative results were drawn from the MATH dataset, which has only true claims as far as I know. It appears that additional qualitative results that include bad claims with coherence factuality are needed.
>
> The premise for our work is that given a language model that can potentially produce false claims, we want to remove such claims without any reliance or access to the gold labels from the MATH dataset. This is implemented through our filtering protocol, which makes use of a frequency-based scoring function, as a robust, grounding-free source of determining correctness for new examples; as such, our method is agnostic of the gold labels (true claims) in the dataset.
>
> **Graph Proxies**
> > Additionally, an approximate deducibility graph is obtained by creating graph proxies using GPT-4o, but this does not provide theoretical guarantee, which is also mentioned in the paper. This paper said that these graph proxies provide a benefit in imposing the property called dependency, but as mentioned above, it does not come as a big advantage if bad claims are considered, so it appears that the theoretical guarantee of conformal prediction is not fully utilized.
>
> The lower bound of Theorem 1 holds regardless of the quality of the graphs, although low quality graphs may harm claim retention. The only caveat is that coherent factuality is obtained relative to the knowledge source on which annotator’s rely on to perform annotation (which, in the context of our work, is solely limited to the annotator’s background knowledge). The upper bound is satisfied by graphs that satisfy Definition 4, and relative to human-annotated “ideal” deducibility graphs, the GPT-4o graphs satisfied this definition (see analysis of this below). Furthermore, as noted in our work, such a graph with additional edges still serves as an approximate deducibility graph per our definition, since we do not require minimality; the presence of bad claims and edges between this is therefore not a concern, as the calibration algorithm produces an appropriate threshold to filter out bad claims.
>
> > As mentioned above, are there any results from experiments using a human annotated ideal graph other than GPT-4o?
> Yes, we also have the following results comparing GPT-4o-generated graphs to human-annotated ideal graphs, and analyzing our method’s performance with these gold graphs (repeated from general response). For GPT-4o, we manually constructed ideal graphs for the first ten examples. The edit distance to the ideal deducibility graph was on average 1.8; the edit distance to an approximate deducibility graph was 0 (meaning each graph considered satisfied Definition 4, which is all that is necessary for both bounds to hold).
>
> Both methods were calibrated, so coherent factuality was approximately the target $1 - \\alpha$ in either case. Thus, we only include retention results in this table. Note that 1.0 because the baseline accuracy here was 70% (no filtering needed at $\\alpha = 0.3$).
>
> | $\\alpha$ | Claim Retention (Human-Generated Graphs) | Claim Retention (GPT-Generated Graphs) |
> | --------- | ---------------------------------------- | -------------------------------------- |
> | 0.1       | 0.33                                     | 0.33                                   |
> | 0.2       | 0.74                                     | 0.86                                   |
> | 0.3       | 1.0                                      | 1.0                                    |
>
>
> The plots of the results which include the realized coherent factuality for each of these settings are in Appendix F.

---

> ### Author Response · Authors · 2024-11-22
> **Response to Reviewer wbML (Part 2/2)**
>
> **Qualitative Examples**
> > Are there extra qualitative results for other datasets (e.g., FELM)?
> Yes, we include a qualitative example from FELM below, as well as a few more in Appendix L.2 of the revised version of our paper.
>
> Question: Jessica makes $\\$2,000.00$ a month.  She sets 25\% of her paycheck aside to put towards fancy shoes.  Each pair of shoes she buys costs $\\$1,000.00$.  How many shoes can she buy in a year?
>
> Independent Factuality:
> Jessica sets aside 25\% of her paycheck, which is: $\\$2,000.00$ x 0.25 = $\\$500.00$
> So Jessica can buy 6 pairs of shoes in a year with the money she sets aside from her paycheck.
>
> Coherent Factuality:
> Jessica sets aside 25\% of her paycheck, which is: $\\$2,000.00$ x 0.25 = $\\$500.00$
> To figure out how many pairs of shoes she can buy in a year, we need to multiply the number of pairs she can buy in a month by 12 (the number of months in a year):$\\$500.00$ x 12 = $\\$6,000.00$.
>
> ---
>
> We hope that this addresses the points raised in the review, and we would be happy to address any concerns that remain!

---

> > ### Comment · Reviewer_wbML · 2024-11-25
> >
> > Thanks for the your responses and for providing corrections on certain points. I have adjusted my score to 6.

---

### Author Response · Authors · 2024-11-22
**General Response (Part 1/2)**

We sincerely thank the reviewers for their valuable feedback, comments, suggestions, and questions. The concerns raised primarily surround the points of the practical utility of coherently factual outputs, the quality of the graph proxies generated by GPT-4o, how bad claims are handled, and how the ground truth is instantiated. We address each of these points below:

1. *Practical utility of coherently factual outputs*: Reviewers DfTB and D79W raise the question of whether the responses generated after filtering are useful, and can improve performance / correctness. In our work, we examined the utility of coherently factual outputs by re-prompting the model conditional on the outputs of our algorithm. This involves generating a coherently factual output through our subgraph filtering protocol (which constitutes a partial reasoning chain), and re-prompting the model to complete the reasoning chain (i.e. filling in any missing steps). The intuition driving this procedure is that a coherent response consisting of subclaims that are directly related to one another in the approximate deducibility graph is easier to complete than a non-coherent response consisting of disjoint claims. These are the results included in Table 1 of our work. Our findings reinforce the notion that while coherent factuality is stricter than independent factuality, our protocol does not result in correctness being sacrificed at the cost of coherence. In fact, it attains both a lower factuality error as well as higher claim retention relative to independent factuality; for $\\alpha = 0.05$, the factuality error was reduced to 0.10 with bootstrapping on coherently factual responses, as opposed to 0.26 when bootstrapping independently factual responses. This appears to validate our hypothesis on coherently factual responses being more amenable to completion by re-prompting. We have included the methodology behind this bootstrapping method in Appendix I.

2. *Quality of the graph proxies*: As reviewers wbML and DfTB note, our work relies on GPT-4o to produce proxy graphs for reasoning problems, though they may be of seemingly unknown quality. First, we note that the conformal lower bound is independent of graph quality and only requires data exchangeability. However, low quality graphs might harm claim retention. To measure the quality of our GPT graph proxies, we manually construct “ideal” graphs for the first ten problems in MATH, such that we have gold (human-annotated) and silver (model-generated) graphs for those samples. To determine proxy-ideal similarity, we compute edit distance. The edit distance from GPT proxies to the ideal deducibility graph was on average 1.8; the edit distance to any approximate deducibility graph was 0 (meaning each graph considered satisfied Definition 4, which is all that is necessary for both bounds to hold). Both methods were calibrated, so coherent factuality was approximately the target $1 - \\alpha$ in either case. Thus, we only include retention results in this table. Note that 1.0 because the baseline accuracy here was 70% (no filtering needed at $\\alpha = 0.3$)


| $\\alpha$ | Claim Retention (Human-Generated Graphs) | Claim Retention (GPT-Generated Graphs) |
| --------- | ---------------------------------------- | -------------------------------------- |
| 0.1       | 0.33                                     | 0.33                                   |
| 0.2       | 0.74                                     | 0.86                                   |
| 0.3       | 1.0                                      | 1.0                                    |


The plots of the results which include the realized coherent factuality for each of these settings are in Appendix F. As noted above, the graph quality is not necessary to obtain the lower bound in Theorem 1, and this is supported by our calibration plots. Nonetheless, we have empirical evidence that the graphs are high-quality -- in fact, as observed through the table above and the plots in Appendix F, they outperform human-annotated ideal graphs (albeit for a small number of samples). This is likely due to the model-generated graphs capturing dependency, which in practice refers to how prior claims are considered in producing subsequent ones, and the non-minimal nature of approximate deducibility graphs. Further evidence of our graphs’ quality is the fact that silver calibration (which assumes validity of the deducibility graph) yields effective gold validation (which does not depend on graph validity).

---

### Author Response · Authors · 2024-11-22
**General Response (Part 2/2)**

3. *Handling of bad claims*: The inclusion of bad claims in the approximate deducibility graph or such claims receiving a high self-consistency score is not a concern because our calibration algorithm ensures that only claims that meet calibrated thresholds over self-consistency score as well as the graphical definition of coherence are included in the final output. Our algorithm dynamically adjusts the threshold to filter out claims with lower scores, such that only those with a high probability of being factual remain. If bad claims are highly self-consistent, the threshold would rise accordingly, imposing stricter conditions for any one claim to be included in the output. Additionally, subgraphs containing bad claims are effectively excluded because their scores, influenced by the bad claims, are likely to fall below the threshold with high probability. This combination of score-based filtering and the representation of dependencies guarantees that the final output maintains correctness and coherence, regardless of the initial presence or consistency of bad claims.

4. *Instantiation of ground truth*: With regards to the use of the ground truth (subset of claims we assume to be valid, denoted $C_{true}$) -- notably, our algorithm does not directly require knowledge of $C_{true}$ or a complete fixed instantiation of the ground truth in practice. Our guarantees do depend on the quality of annotations in the calibration set, which are with respect to $C_{true}$. To this effect, in the annotation phase, we relied on the annotators’ understanding of the required prior knowledge and the given context of the problem to serve as a reasonable proxy, as in Mohri and Hashimoto, 2024; for math problems, however, we reasonably assume that the annotators’ conceptions will be uniform, with similar levels of mathematical knowledge and similar conceptions of mathematical substantiation.


### Update to Paper
We have made the following updates to our paper in the revised draft based on the valuable feedback of the reviewers; our paper has been uploaded above (changes are visible in teal):
* A clarification of the minimal set of claims, in Section 3.1.
* An explanation of the self-consistency scoring function used in practice, in Section 4.
* Added experiments with Llama-3.1-70-Instruct, an open-source model, in Appendix E, and clarified that responses were generated from GPT-4 in the experiments included in the main text.
* Updated Table 1 and R5 in Section 5 to address the iterative application of our algorithm and its implications in boosting correctness and the utility of outputs.
* An analysis of claim retention using ten human-annotated “ideal” deducibility graphs, as opposed to using ten GPT-4o-generated “approximate” deducibility graphs, in Appendix F.
* Added our cost estimates to Appendix J, for reproducibility.
* Added example outputs from FELM to Appendix L.2, comparing the behavior of independent factuality and coherent factuality.

---

### Meta-Review · Area_Chair_dk84 · 2024-12-23

**Metareview:**

The paper introduces a conformal prediction framework tailored for reasoning tasks in LMs. It addresses limitations of prior methods by proposing "coherent factuality," capturing correctness across interconnected claims rather than independently evaluating them. The methodology employs deducibility graphs, combining graph-based claim scoring with a split conformal prediction approach. Experiments on the MATH and FELM datasets demonstrate the method's ability to improve factuality retention while maintaining high correctness.

*Strengths*:
-The paper proposes an innovative "coherent factuality" approach, extending conformal prediction frameworks to reasoning tasks, which have unique dependencies.
-The method achieves substantial improvements in correctness without sacrificing coherence.
-The framework is adaptable to multiple language models, including open-source options like Llama.
-It examines the quality of deducibility graphs, compares proxy and human-annotated graphs, and validates its calibration guarantees.

*Weaknesses*:
-Reviewers raised concerns about inconsistent terminology and unclear descriptions, although these were addressed in the revised version.
-Several reviewers noted issues regarding how the "ground truth" was instantiated in the experiments.
-There were limited direct evaluations of how filtering outputs impacts the utility in solving full reasoning problems.
-Potential for accepting bad (incorrect) claims just because they are consistent with the rest of the claims.
-Reliance on GPT-generated graphs without a formal guarantee was noted as a limitation, albeit mitigated by empirical validation.

While reviewers all indicated marginally above acceptance rate, none expressed strong enthusiasm for the work. This may have been driven more by the paper's solid technical contributions and execution than by its ability to excite or inspire transformative potential. While the authors added experiments with Llama, reviewers might still find the limited dataset scope (MATH and FELM) and absence of additional task types a concern.

**Additional Comments On Reviewer Discussion:**

The rebuttal period highlighted several key points:
-Reviewers questioned the reliability of GPT-4o-generated graphs. Authors provided empirical evidence comparing these graphs with human-annotated ones.
-Regarding instantiation of ground truth, the authors partially addressed this by clarifying that their method does not require complete ground truth knowledge
-Direct evaluation of downstream tasks was limited, but the authors introduced metrics like legibility and correctness retention, which showed their method's utility.
-The authors addressed concerns with revisions clarifying terminology and methodology, particularly in sections on graph construction and scoring functions.
-New experiments with Llama-3.1-70B-Instruct provided additional validation, addressing concerns about the reproducibility and applicability of the method across models.
-The reviewers largely appreciated the responses, raising scores in acknowledgment of the revisions and additional analyses.

---

### Decision · Program_Chairs · 2025-01-22

Accept (Poster)